Multigene phylogeny of the scyphozoan jellyfish family Pelagiidae reveals that the common U.S. Atlantic sea nettle comprises two distinct species (Chrysaora quinquecirrha and C. chesapeakei)

Bayha Keith M. 1 2 bayhak@si.edu
http://orcid.org/0000-0002-3664-9691 Collins Allen G. 3
Gaffney Patrick M. 4
1 Department of Invertebrate Zoology, Smithsonian Institution, National Museum of Natural History , Washington, DC , USA
2 Department of Biological Sciences, University of Delaware , Newark, DE , USA
3 National Systematics Laboratory of NOAA’s Fisheries Service, Smithsonian Institution , Washington, DC , USA
4 College of Earth, Ocean and Environment, University of Delaware , Lewes, DE , USA
Toonen Robert
Electronic publication date: 2017 Oct 13
Publication date: 2017
Volume: 5
Electronic Location ID: e3863
Received 2017 Jun 12; Accepted 2017 Sep 8
Copyright: © 2017 Bayha et al.
Copyright year: 2017
Copyright holder: Bayha et al.
License: This is an open access article distributed under the terms of the Creative Commons Attribution License, which permits unrestricted use, distribution, reproduction and adaptation in any medium and for any purpose provided that it is properly attributed. For attribution, the original author(s), title, publication source (PeerJ) and either DOI or URL of the article must be cited.
License URL: https://creativecommons.org/licenses/by/4.0/

Keywords: Evolution, Phylogeny, Jellyfish, Chrysaora, Sea nettle, Scyphozoa, Cryptic species

Funding: Lerner-Gray Grant for Marine Research (American Museum of Natural History) This work was supported by Lerner-Gray Grant for Marine Research (American Museum of Natural History) to Keith M. Bayha. There was no additional external funding received for this study. The funders had no role in study design, data collection and analysis, decision to publish, or preparation of the manuscript.

==============================
Background

Species of the scyphozoan family Pelagiidae (e.g., Pelagia noctiluca, Chrysaora quinquecirrha) are well-known for impacting fisheries, aquaculture, and tourism, especially for the painful sting they can inflict on swimmers. However, historical taxonomic uncertainty at the genus (e.g., new genus Mawia) and species levels hinders progress in studying their biology and evolutionary adaptations that make them nuisance species, as well as ability to understand and/or mitigate their ecological and economic impacts.

Methods

We collected nuclear (28S rDNA) and mitochondrial (cytochrome c oxidase I and 16S rDNA) sequence data from individuals of all four pelagiid genera, including 11 of 13 currently recognized species of Chrysaora. To examine species boundaries in the U.S. Atlantic sea nettle Chrysaora quinquecirrha, specimens were included from its entire range along the U.S. Atlantic and Gulf of Mexico coasts, with representatives also examined morphologically (macromorphology and cnidome).

Results

Phylogenetic analyses show that the genus Chrysaora is paraphyletic with respect to other pelagiid genera. In combined analyses, Mawia, sampled from the coast of Senegal, is most closely related to Sanderia malayensis, and Pelagia forms a close relationship to a clade of Pacific Chrysaora species (Chrysaora achlyos, Chrysaora colorata, Chrysaora fuscescens, and Chrysaora melanaster). Chrysaora quinquecirrha is polyphyletic, with one clade from the U.S. coastal Atlantic and another in U.S. Atlantic estuaries and Gulf of Mexico. These genetic differences are reflected in morphology, e.g., tentacle and lappet number, oral arm length, and nematocyst dimensions. Caribbean sea nettles (Jamaica and Panama) are genetically similar to the U.S. Atlantic estuaries and Gulf of Mexico clade of Chrysaora quinquecirrha.

Discussion

Our phylogenetic hypothesis for Pelagiidae contradicts current generic definitions, revealing major disagreements between DNA-based and morphology-based phylogenies. A paraphyletic Chrysaora raises systematic questions at the genus level for Pelagiidae; accepting the validity of the recently erected genus Mawia, as well as past genera, will require the creation of additional pelagiid genera. Historical review of the species-delineating genetic and morphological differences indicates that Chrysaora quinquecirrha Desor 1848 applies to the U.S. Coastal Atlantic Chrysaora species (U.S. Atlantic sea nettle), while the name C. chesapeakei Papenfuss 1936 applies to the U.S. Atlantic estuarine and Gulf of Mexico Chrysaora species (Atlantic bay nettle). We provide a detailed redescription, with designation of a neotype for Chrysaora chesapeakei, and clarify the description of Chrysaora quinquecirrha. Since Caribbean Chrysaora are genetically similar to Chrysaora chesapeakei, we provisionally term them Chrysaora c.f. chesapeakei. The presence of Mawia benovici off the coast of Western Africa provides a potential source region for jellyfish introduced into the Adriatic Sea in 2013.

Introduction

Scyphozoan jellyfishes (Cnidaria, class Scyphozoa), which include the conspicuous moon, lion’s mane and sea nettle jellyfishes, exhibit significant and widespread economic and ecological impacts on a wide array of marine and estuarine communities. Jellyfish aggregations, blooms, and swarms damage economically important fisheries, close tourist beaches by stinging swimmers, clog intakes of coastal power and desalination plants, invade ecosystems, and can affect oxygen levels when mass numbers of carcasses are deposited (Arai, 1997; Purcell, Uye & Lo, 2007; Richardson et al., 2009; Bayha & Graham, 2014; Qu et al., 2015). On the other hand, jellyfish serve important roles as major prey items for some fish and sea turtles, in carbon capture and advection to the Deep Ocean, as important microhabitat for fish, invertebrates, and symbiotic algae, and as economic resources for humans (as food and therapeutic compounds) (Doyle et al., 2014, Omori & Nakano, 2001; Castro, Santiago & Santana-Ortega, 2002; Arai, 2005; Houghton et al., 2006; Lynam & Brierley, 2007; Ohta et al., 2009; Lebrato et al., 2012; Diaz Briz et al., 2017). Recent attention given to large medusae blooms has led to speculation that anthropogenic events are driving global increases in jellyfish bloom magnitudes, though long-term data sets are still equivocal on this point (Richardson et al., 2009; Brotz & Pauly, 2012; Condon et al., 2013).

Despite their importance, evolutionary and taxonomic relationships of even some of the most recognizable scyphozoan species remain unsettled, which can impede our abilities to effectively study, predict and mitigate the ecological and economic effects of these nuisance species. Recent systematics studies have directly challenged taxonomic relationships at all levels. A mitogenomic analysis recently challenged the placement of the order Coronatae, such as Periphylla, within Scyphozoa (Kayal et al., 2013; but see Kayal et al., 2017) and the new family Drymonematidae was created based on morphological, molecular, and life history data (Bayha & Dawson, 2010; Bayha et al., 2010). Studies employing molecular and/or morphological data have revealed novel species in multiple scyphozoan genera, including the moon jellyfish Aurelia (Dawson & Jacobs, 2001; Schroth et al., 2002; Dawson, 2003), the genus Drymonema (Bayha & Dawson, 2010), the upside down jellyfish Cassiopea (Holland et al., 2004), and the lion’s mane jellyfish Cyanea (Dawson, 2005; Kolbasova et al., 2015). Many of these studies have uncovered unrecognized jellyfish invasions and clarified evolutionary relationships in the group (from order to species level) vital to understanding their ecological and economic impacts, and elucidating the evolution of traits that permit these impacts.

The scyphozoan family Pelagiidae (Gegenbaur, 1856), currently made up of four genera (Pelagia, Chrysaora, Sanderia, and Mawia), contains some of the world’s most notorious blooming jellyfish. The geographically widespread mauve stinger (Pelagia noctiluca) forms dense aggregations that heavily impact aquaculture, fisheries, and tourism along the North Sea and Mediterranean Sea (Canepa et al., 2014). Recently, a species found for the first time in the Mediterranean was described and assigned first to the genus Pelagia (Piraino et al., 2014), but later to the novel genus Mawia, based on molecular and morphological data (Avian et al., 2016). Blooms of the jellyfish Chrysaora fulgida (previously identified as Chrysaora hysoscella) have increased over past decades in the Northern Benguela current on the west coast of Africa, coinciding with decreased fish catches and general breakdown of beneficial trophic interactions as compared to nearby ecosystems not jellyfish-dominated (Lynam et al., 2006; Flynn et al., 2012; Roux et al., 2013). Likewise, blooms of very large Chrysaora plocamia medusae form off the coast of Peru, interfering with fisheries, aquaculture, and power plants by clogging nets, seines, and water intakes (Mianzan et al., 2014).

A species of special note is the U.S. Atlantic sea nettle Chrysaora quinquecirrha (Desor, 1848), one of the most recognizable, well-studied, and ecologically important jellyfish along the U.S. Atlantic and Gulf of Mexico coasts (Mayer, 1910; Hedgepeth, 1954; Larson, 1976). Because its predation pressure shows ecosystem-wide, controlling influence on zooplankton dynamics (Feigenbaum & Kelly, 1984; Purcell, 1992; Purcell & Decker, 2005), Chrysaora quinquecirrha has been termed a keystone predator for the Chesapeake Bay ecosystem (Purcell & Decker, 2005). The jellyfish negatively impacts economically important fisheries by feeding on eggs and larvae (Duffy, Epifanio & Fuiman, 1997; Purcell, 1997) and blooms impact tourism by stinging swimmers (Cargo & Schultz, 1966; Schultz & Cargo, 1969; Cargo & King, 1990). As a result, a program was developed to predict both real-time occurrences of sea nettle blooms (Decker et al., 2007) and year-to-year bloom magnitudes using past data on environmental conditions (salinity, temperature, etc.) that favor jellyfish populations (Purcell et al., 1999; Purcell & Decker, 2005).

Generic definitions within what is currently accepted as family Pelagiidae (Gegenbaur, 1856) have been historically vague and genera have traditionally been differentiated, to a great extent, on a single morphological character (tentacle number). The generic names Pelagia and Chrysaora were originated by Péron & Lesueur (1810), though both included species not recognized today as pelagiids. Gegenbaur (1856) was the first to create a higher taxon, the family Pelagiidae, including all pelagiids known at the time, but which also included some jellyfish currently classified as coronates. Noting differences based on tentacle number between Chrysaora and Pelagia, Agassiz (1862) erected a new genus, Dactylometra, within the family. Among other characters, Agassiz (1862) classified genera based on tentacle and lappet numbers: Pelagia (eight tentacles, 16 marginal lappets), Chrysaora (24 tentacles, 32 marginal lappets), and Dactylometra (40 tentacles, 48 marginal lappets). Kishinouye (1902) subsequently described the genus Kuragea (56 tentacles, 64 marginal lappets) and Goette (1886) described Sanderia (16 tentacles, 32 lappets, and 16 rhopalia). To the genus Dactylometra, Agassiz (1862) added Pelagia quinquecirrha (Desor, 1848) from Nantucket Bay (MA) and Chrysaora lactea (Eschscholtz, 1829) from Rio de Janeiro. Based on established generic definitions, Piraino et al. (2014) placed an undescribed, presumably non-indigenous Mediterranean pelagiid, Pelagia benovici, in the genus Pelagia. However, Avian et al. (2016) created the novel genus Mawia for this new species (Mawia benovici) based on fine-scale morphological characters (tentacle, gonad, and basal pillar morphology) and molecular differences from other pelagiid genera included in a lightly sampled phylogenetic analysis of Pelagiidae.

Not long after Agassiz erected Dactylometra, Dactylometra quinquecirrha served to cast doubt on pelagiid generic discrimination. Bigelow (1880) recognized that some brackish water (e.g., Chesapeake Bay) Dactylometra quinquecirrha matured at 24 tentacles (a character of Chrysaora) rather than 40 (a character of Dactylometra), something Mayer (1910), saw as the “Chrysaora” stage in their development to the “Dactylometra” stage. Stiasny (1930) also cast doubt on the ability to effectively differentiate Chrysaora and Dactylometra. As a result, Kramp (1955) reasoned Dactylometra and Kuragea to be merely developmental stages and subsumed both within the genus Chrysaora (Eschscholtz, 1829), since it has taxonomic priority. Calder (1972) determined that Chrysaora quinquecirrha went through stages of one to more than seven tentacles per octant, often in the same geographic region, supporting the contentions of Mayer (1910) and Kramp (1955). A morphology-based phylogeny of the Pelagiidae (Gershwin & Collins, 2002) indicated two groups coinciding with the previous genera Dactylometra and Chrysaora, but noted that the weak phylogenetic support would make resurrecting the genus Dactylometra premature. Another morphology-based phylogeny (Morandini & Marques, 2010) found support for a Dactylometra clade based on tentacle and lappet number, but noted that this would require many Chrysaora species to have their own genera. A robust phylogenetic hypothesis of relationships within Pelagiidae based on comprehensive taxon sampling is an important step toward removing taxonomic confusion at the genus and species-levels, including assessing the taxonomic status of the new genus Mawia (Avian et al., 2016) and clarifying taxonomic questions related to Chrysaora quinquecirrha.

In order to examine evolutionary relationships and taxonomic boundaries in the family Pelagiidae, with special focus on the genus Chrysaora and the species Chrysaora quinquecirrha, we collected nuclear (large subunit ribosomal rDNA) and mitochondrial (cytochrome c oxidase I and large subunit ribosomal rDNA) sequence data from individuals representing all four extant genera (Chrysaora, Mawia, Pelagia, and Sanderia), including 11 currently recognized species of Chrysaora and one species each of Mawia (Mawia benovici), Pelagia (P. noctiluca), and Sanderia (S. malayensis). To further examine the taxonomy of the U.S. Atlantic sea nettle Chrysaora quinquecirrha, specimens were included from its entire range along the U.S. Atlantic and Gulf of Mexico coasts (estuarine and coastal), taking care to sample all recognized morphotypes, with representatives also examined morphologically (macromorphology and cnidome).

Materials and Methods

Sample collection

Specimens were collected in the field or at public aquaria husbandry facilities, either by the authors or others with extensive knowledge of Scyphozoa, in an effort to collect as many species of Chrysaora as possible, as well as representative species of Pelagia, Mawia, and Sanderia (Table 1; Fig. 1). An unknown and unidentified pelagiid specimen was collected from Dakar, Senegal and was accompanied by a photograph that did not allow for specific identification (Fig. S1). For Chrysaora quinquecirrha, samples were collected from 10 different sites along the Atlantic and Gulf of Mexico coasts (Table 1; Fig. 2), covering both coastal and estuarine environments, with the intention of capturing as many structural and color morphotypes as possible (Fig. 3). Both white (Table 1: NF1–NF3) and red-striped (Table 1: NF4–NF5) color morphs (Figs. 3C and 3D) were collected from Norfolk, VA (NF). In all cases, a small piece of gonad, tentacle or oral arm tissue was excised and preserved in 80–99% ethanol or DMSO-NaCl solution (Dawson, Raskoff & Jacobs, 1998). Where possible for some sites (Table S1), individuals were also preserved in 4% buffered formalin and seawater for later morphological analyses. Additional published pelagiid sequences were included in the final data set (Table 2).

Table 1 Geographic source regions of samples used for molecular analyses in this study, identified by taxon (original, morphologically based identification) and molecular ID (identification after molecular analyses).

Original ID	Final ID	Location	Code	n	
COI	16S	28S	
Chrysaora achlyos	C. achlyos	Monterey Bay Aquarium*	MBA	1	1	1	
Chrysaora africana	C. africana	Coastal Namibia	NAM	2	2	2	
Chrysaora chinensis	C. chinensis	Monterey Bay Aquarium^	MBA	2	2	2	
Chrysaora colorata	C. colorata	Aquarium of the Americas+	AQA	1	1	1	
Chrysaora fulgida	C. fulgida	Coastal Namibia	NAM	5	5	2	
Chrysaora fuscescens	C. fuscescens	Aquarium of the Americas+	AQA	1	1	HM194868	
Chrysaora hysoscella	C. hysoscella	Cork, Ireland	IRE	3	3	3	
Chrysaora lactea	Chrysaora c.f. chesapeakei	Kingston, Jamaica	JAM	5	5	2	
Chrysaora lactea	C. lactea	Rio de la Plata, Argentina	ARG	1	1	1	
Chrysaora melanaster	C. melanaster	Bering Sea	BER	–	1	AY920780	
Chrysaora melanaster	C. pacifica	Monterey Bay Aquarium	MBA	1	1	HM194864	
Chrysaora plocamia	C. plocamia	Puerto Madryn, Argentina	PMA	2	2	2	
Chrysaora quinquecirrha	C. quinquecirrha	Buzzard’s Bay, MA (USA)	MA	1	1	1	
Chrysaora quinquecirrha	C. quinquecirrha	Cape Henlopen, DE (USA)	CHP	3	3	2	
Chrysaora quinquecirrha	C. quinquecirrha	Offshore South Carolina (USA) (32.60 N, 79.21 W)	OSC	2	2	1	
Chrysaora quinquecirrha	C. chesapeakei	Charlestown Pond, RI (USA)	RI	4	4	–	
Chrysaora quinquecirrha	C. chesapeakei	Tom’s River Harbor, NJ (USA)	NJ	3	3	1	
Chrysaora quinquecirrha	C. chesapeakei	Rehoboth Bay, DE (USA)	RB	3	3	–	
Chrysaora quinquecirrha	C. chesapeakei	Norfolk, VA (USA)	NF	5	5	–	
Chrysaora quinquecirrha	C. chesapeakei	Pamlico Sound, NS (USA)	PAM	3	3	–	
Chrysaora quinquecirrha	C. chesapeakei	St. Simon’s Island, GA (USA)	GA	3	3	1	
Chrysaora quinquecirrha	C. chesapeakei	Perdido Pass, AL (USA)	AL	3	3	1	
Pelagia noctiluca	P. noctiluca	Offshore Virginia (USA) (37.81 N, 73.91 W)	OVA	1	1	HM194865	
Sanderia malayensis	S. malayensis	Monterey Bay Aquarium	MBA	1	1	HM194861	
Unknown Pelagiidae	M. benovici	Dakar, Senegal	SEN	2	2	1	
Cyanea capillata	C. capillata	Blomsterdalen, Norway	BLO	1	1	HM194873	
Notes:

For six individuals, 28S sequences from those individuals were published previously. For S. malayensis, 16S/COI and 28S sequences came from the same culture, but two different individuals. For some aquarium specimens, the geographic source region for the culture is known: *near Los Angeles, CA (USA); ^Northern Malaysia; +near Monterey Bay, CA (USA).

Figure 1 World map showing collecting sites of animals sequenced for this study.

Final species designations are employed. All aquarium samples (Chrysaora achlyos, Chrysaora chinensis, Chrysaora colorata, Chrysaora fuscescens, and Chrysaora pacifica) originated from cultures at the Monterey Bay Aquarium, although some were obtained from the Aquarium of the Americas. Their locations on the map are based on original collection locations for the aquarium cultures (W. Patry, 2015, personal communication).

Figure 2 Collection locations of Chrysaora quinquecirrha s.l. medusae used in this study.

Abbreviations all refer to Table 1 and Table S1. (A–C) are enlargements of rectangular inset regions. The star at Nantucket harbor indicates the type locality of C. quinquecirrha (Desor, 1848). Diamonds represent important museum collection sites (Table S1). Site RI is within the enclosed Charlestown Pond, RI (41.364.765 N, 71.628865 W). Site NJ is at Ocean Gate Yacht Club (39.930490 N, 74.140448 W) on Toms River, inside Barnegat Bay. Site RB was collected from inside Rehoboth Bay, DE (38.688091 N, 75.077114 W). All Chesapeake Bay samples (NF and Gloucester Point, VA) were taken from well within the Chesapeake Bay. Site PAM was collected from Engelhard, NC (35.509102 N, 75.989712 W), well within Pamlico Sound. CST was taken from within Charleston Harbor (32.786995 N, 79.909297 W). Site GA was taken from Fancy Bluff Creek, upstream from Saint Simons Sound, GA (31.166291 N, 81.416032 W). Sample sites with individuals finally designated as Chrysaora quinquecirrha are in white and those with individuals finally designated as Chrysaora chesapeakei in black.

Figure 3 Various morphs of Chrysaora quinquecirrha s.l.

(A) Offshore South Carolina (OSC); (B) Sample taken from offshore Georgia; (C) Engelhard, NC (PAM); (D) White Chesapeake Bay color morph (Broome’s Island, MD—Patuxent River); (E) Red-striped Chesapeake Bay color morph (Solomons, MD—Patuxent River). Note that medusae from (A) to (B) have five tentacles per octant, while (C)–(E) have three tentacles per octant. Medusae in (A, C) were included in this study’s phylogenetic analyses. (A: OSC1; C: PAM1). (A, B) represent individuals finally designated as Chrysaora quinquecirrha; (C–E) represent individuals finally designated as Chrysaora chesapeakei. Photo Credits: (A) Shannon Howard; (B) Greg McFall-NOAA; (E) Robert Condon.

Table 2 Geographic source regions of previously published sequences used in in this study identified by taxon (previous identification) and molecular ID (identification after molecular analyses).

Original ID	Final ID	Location	Code	n	
COI	16S	28S	
Chrysaora melanaster	C. melanaster	Bering Sea	BER1	KJ026191	–	–	
Chrysaora melanaster	C. melanaster	Bering Sea	BER2	KJ026212	–	–	
Chrysaora melanaster	C. melanaster	Bering Sea	BER3	KJ026256	–	–	
Chrysaora sp.	Chrysaora c.f. chesapeakei	Bocas del Toro, Panama	PAN	JN700941*	JN700941*	AY920779*	
Chrysaora pacifica	Chrysaora pacifica	Kyoto, Japan	KYO	LC191577	–	–	
Chrysaora quinquecirrha	C. pacifica	Geoje-do, Korea	KOR	HQ694730	HQ694730	–	
Chrysaora sp.	Chrysaora sp. 1	Noosa Heads, Australia	AUS	DQ083524	–	–	
Chrysaora sp.	C. chinensis	Malaysia	MAL1	–	JN184784	–	
Chrysaora sp.	C. chinensis	Malaysia	MAL2	–	JN184785	–	
Chrysaora sp.	C. chinensis	Malaysia	MAL3	–	JN184786	–	
Pelagia benovici	P. benovici	Northern Adriatic Sea	ADR1	KJ573409	–	KJ573396	
Pelagia benovici	P. benovici	Northern Adriatic Sea	ADR2	KJ573410	–	KJ573397	
Pelagia benovici	P. benovici	Northern Adriatic Sea	ADR3	KJ573412	–	KJ573401	
Pelagia noctiluca	P. noctiluca	Southern Tyrrhenian Sea, Italy	TYR	KJ573419	–	KJ573408	
Pelagia noctiluca	P. noctiluca	Cape Town, South Africa	SA	JQ697961	–	–	
Pelagia noctiluca	P. noctiluca	Dispensa Island, Costa Rica	CR1	JX235441	–	–	
Pelagia noctiluca	P. noctiluca	Dispensa Island, Costa Rica	CR2	–	JX235404	–	
Pelagia noctiluca	P. noctiluca	Dispensa Island, Costa Rica	CR3	–	JX235405	–	
Pelagia c.f. panopyra	Pelagia c.f. panopyra	Papua, New Guinea	PNG	KJ573422	–	–	
Note:

* Sequences came from the same individual.

DNA extraction, PCR amplification and DNA sequencing

Genomic DNA was extracted from preserved tissue samples by CTAB (cetyltrimethylammonium bromide) methods (Ausubel et al., 1989) and stored at −20 °C. Polymerase chain reaction (PCR) amplifications targeted three genetic regions: mitochondrial large subunit ribosomal DNA (16S), cytochrome c oxidase subunit I (COI), and nuclear large subunit ribosomal DNA (28S) using primers shown in Table S2. We chose genetic regions that have been useful in examining species boundaries and/or examining genus and family level relationships in the Scyphozoa (Dawson & Jacobs, 2001; Schroth et al., 2002; Holland et al., 2004; Dawson, 2005; Dawson, Gupta & England, 2005; Bayha & Dawson, 2010). Reaction conditions for 16S consisted of one cycle of 94 °C for 180 s, then 38 cycles of 94 °C for 45 s, 50 °C for 60 s, and 72 °C for 75 s, followed by a final step of 72 °C for 600 s and storage at 4 °C. Reaction conditions for COI consisted of one cycle of 94 °C for 180 s, followed by two cycles of 94 °C for 45 s, 46 °C for 60 s, and 72 °C for 75 s, two cycles of 94 °C for 45 s, 47 °C for 60 s, and 72 °C for 75 s, and 35 cycles of 94 °C for 45 s, 48 °C for 60 s, and 72 °C for 75 s, followed by a final step of 72 °C for 600 s and storage at 4 °C. Lastly, reactions conditions for 28S consisted of 94 °C for 180 s, then 38 cycles of 94 °C for 45 s, 48 °C for 60 s, and 72 °C for 90 s, followed by 72 °C for 600 s then storage at 4 °C. Successful amplification was evaluated by running the PCR products on a 2% agarose gel. PCR amplicons were directly sequenced using a combination of sequencing primers (Table S2). DNA sequencing was performed by University of Washington High Throughput Genomics Unit (Seattle, WA, USA) or Beckman-Coulter Genomics (Danvers, MA, USA). Sequences were assembled using Lasergene SeqMan Pro v. 8.1.5 (DNAStar, Inc., Madison, WI, USA) and then compared to the GenBank nucleotide database using BLASTn or BLASTx (Altschul et al., 1997) to confirm identity of sequenced region and ensure no sequencing errors that affected amino acid reading frames (COI). All DNA sequences were submitted to NCBI GenBank (MF141552–MF141593; MF141595–MF141646; MF141648–MF141718; MF167556–MF167568).

Phylogenetic reconstruction

For all analyses, Cyanea capillata (Blomsterdalen, Norway) was used as the outgroup because it was shown to be among those scyphozoans least diverged from Pelagiidae (Bayha et al., 2010). COI sequences were aligned using CLUSTALX v2.1 (Larkin et al., 2007) under default parameters, and checked by eye using their amino acid translations as a guide. 16S and 28S sequences were aligned using MAAFT v7.245 employing the E-INS-I strategy (Katoh & Standley, 2013), since this strategy has been demonstrated to be effective for loci containing conserved motifs embedded within hypervariable regions (Katoh & Toh, 2008). Hypervariable regions of questionable alignment were removed from the MAAFT alignments using GBlocks v0.91b (Castresana, 2000) under default parameters, except that gapped positions were set to half.

Phylogenetic analyses were run under maximum likelihood (ML) and Bayesian inference (BI) frameworks for COI, 16S, 28S, and a combined dataset. ML phylogenetic trees were constructed using PhyML v3.0 (Guindon et al., 2010), employing the best-fit substitution models assessed using jModelTest v2.1.7 (Darriba et al., 2012) under Akaike (AIC) and Bayesian (BIC) information criteria, as well as decision theory performance-based selection (DT). For COI (TPMμf+I+G), 16S (TIM2+I+G), and combined (GTR+I+G) datasets, selection criteria were unanimous, while BIC and DT chose TrNef+I+G for 28S. A 1,000 bootstrap replicate analysis was performed in PhyML to obtain node support values. BI of gene phylogenies was carried out using MrBayes v3.2.6 (Ronquist et al., 2012). The same model of nucleotide evolution (GTR+I+G, with gamma distribution approximated by four discrete categories) was assumed for all analyses, since it is not possible to implement the less complicated models used in the ML tree searches (in the cases of 16S and COI). For each dataset, two independent MCMC runs were conducted until the standard deviation of split frequencies decreased to less than 0.01 (16S: 6,481,000; COI: 19,608,000; 28S: 1,390,000; combined: 1,002,000) generations, sampling every 1,000. The number of generations was determined by assessment of convergence using the minimum estimated sample size and potential scale reduction factor, as implemented in MrBayes. Posterior probabilities were calculated using all trees other than the first 25%, which were discarded as “burnin”. All trees were visualized using Figtree v1.4.2 (Rambaut, 2014) and redrawn for presentation using Adobe Illustrator CC v19.1.0 (Adobe Systems, Inc., San Jose, CA, USA). Mean interclade and intraclade, as well as minimum interclade sequence divergence values (Kimura 2-parameter) were determined using MEGAv7.0.14 (Kumar, Stecher & Tamura, 2016) and nucleotide statistics calculated in Seaview v4.6 (Gouy, Guindon & Gascuel, 2010).

Morphological analysis of Chrysaora quinquecirrha

While our study did not include a family-wide morphological analysis, we did perform morphological analyses on jellyfish identified as Chrysaora quinquecirrha from the U.S. Atlantic and Gulf of Mexico coasts. We examined a total of 57 formalin-preserved samples we collected from Charlestown Pond (RI), Cape Henlopen (DE), Rehoboth Bay (DE), York River (VA), Charleston (SC), and Dauphin Island (AL) (Table S1). In addition, we examined a total of 63 individuals housed at the Smithsonian Institution National Museum of Natural History (USNM) that were collected from the U.S. Atlantic and Gulf of Mexico coasts and identified as Chrysaora quinquecirrha or Chrysaora sp. (Table S1). We examined morphological characters (and their states) previously employed for Pelagiidae (Gershwin & Collins, 2002) that pertained to the medusa stage, with the addition of maximum oral arm length, where preservation state allowed for its measurement (Table 3). In addition, a total of 35 individuals that were examined morphologically, but not included in the phylogenetic analyses, were assigned to molecular species/clades using mitochondrial 16S sequence data collected using the established procedure described above (Table S1).

Table 3 Morphological characters examined for this study.

Character	Chrysaora quinquecirrha	Chrysaora chesapeakei	
Macromorphology	
Bell diameter (average/median)	114 mm (59–176 mm)	62.2 mm (17–175 mm)	
Tentacles/octant (average ± 95% CI)	5.28 ± 0.45	3.07 ± 0.07	
Tentacles/octant (range)	4.5–6.75	2.75–3.43*	
Lappets/octant (average ± 95% CI)	6.26 ± 0.46	4.08 ± 0.06	
Lappets/octant (range)	5.5–7.75	3.75–4.8	
Maximum oral arm length (average ± 95% CI)	1.24 ± 0.27 times BD	3.00 ± 0.39 times BD	
Maximum oral arm length (range)	0.68–1.81 times BD	1.21–5.58 times BD	
Lappets in size classes	Yes, rhopalar lappets larger	No, lappets of similar size	
Rhopalia number	8	8	
Rhopaliar pits	Deep	Deep	
Septa shape	Bent	Bent	
Septa termination	Near tentacle	Near tentacle	
Spiral oral arms?	No	No	
Manubrium length	Elongated	Elongated	
Manubrium mass	Light	Light	
Warts/papillae	Inconspicuous	Inconspicuous	
Maximum bell diameter	<20 cm^	<20 cm^	
Bell mass	Light	Light	
Dominant color	White, colorless	Variable, white, colorless or red/brown bell	
Exumbrellar marks	Minor bell marks in some cases	Variable, red or brown star shape conspicuous in some cases	
Oral arm color	None	Variable, oral arms can be colored red/brown	
Quadralinga	None	None	
Gonads in pouch?	Yes	Yes	
Gonad shape	Not finger-like	Not finger-like	
Cnidome	
A isorhiza—length vs. width (avg)	20.25 ± 0.38 × 11.27 ± 0.37 μm	26.21 ± 0.50 × 19.74 ± 0.55 μm	
A isorhiza—length vs. width (range)	15.01–22.9 × 9.07–13.16 μm	20.54–33.79 × 15.03–29.77 μm	
a isorhiza—length vs. width (avg)	8.27 ± 0.19 × 4.22 ± 0.07 μm	7.88 ± 0.13 × 4.29 ± 0.07 μm	
a isorhiza—length vs. width (range)	6.56–9.77 × 3.65–4.95 μm	6.32–9.9 × 3.59–5.46 μm	
O isorhiza—length vs. width (avg)	21.64 ± 0.38 × 18.92 ± 0.77 μm	23.10 ± 0.43 × 20.75 ± 0.62 μm	
O isorhiza—length vs. width (range)	17.64–23.97 × 16.08–21.74 μm	17.88–27.51 × 16.07–24.75 μm	
Birhopaloids—length vs. width (avg)	13.58 ± 0.19 × 8.09 ± 0.09 μm	12.73 ± 0.22 × 8.29 ± 0.13 μm	
Birhopaloids—length vs. width (range)	12.31–14.86 × 6.96–8.90 μm	10.96–15.27 × 7.1–10.23 μm	
Notes:

Characters in bold are species diagnostic. All macromorpholgical characters and character states (except maximum oral arm length) are taken from Gershwin & Collins (2002). Cnidome terminology is taken from Morandini & Marques (2010), with average examples in Fig. 8C and Fig. S1.

* If two outlier specimens are included, the upper range is six tentacles/octant.

^ Although maximum bell diameter for Chrysaora quinquecirrha has been recorded as great as 40 mm (Gershwin & Collins, 2002; Morandini & Marques, 2010), no animals >20 mm were observed in this study.

Cnidome of Chrysaora quinquecirrha

Lastly, we examined the cnidome of multiple specimens originally identified as Chrysaora quinquecirrha to determine if species could be delineated based on nematocyst measurements (of each type) and/or nematocyst diversity (counts of nematocyst types). Nematocyst terminology followed convention used in previous studies (Weill, 1934; Calder, 1971; Calder, 1974a; Östman & Hydman, 1997; Morandini & Marques, 2010) in defining four different nematocyst types: holotrichous A-isorhiza, holotrichous a-isorhiza, holotrichous O-isorhiza, and heterotrichous microbasic rhopaloid. In agreement with Morandini & Marques (2010), we use the term heterotrichous microbasic rhopaloid, recognizing that there are likely at least two nematocysts that cannot be effectively delineated based on basic light microscopy, as shown in other previous work (Sutton & Burnett, 1969).

In all cases, formalin-preserved tentacle tissue was homogenized in distilled water in 1.5 mL microcentrifuge tubes and nematocysts were examined using differential interference contrast microscopy (DIC). A small piece of formalin-fixed tentacle tissue was homogenized in 100 μL of distilled water in a 1.5 μL tube using a plastic microcentrifuge pestle until little visible intact tissue remained. A small drop was then placed on a slide under cover slip and examined at 60× in DIC using an Olympus BX63 microscope, with photographs taken using an Olympus DP80 camera run by CellSens Dimension 1.13 (Olympus Life Science, Inc., Waltham, MA, USA).

A total of 15 individuals were examined for nematocyst size measurements (Table S1). In all cases, 10 samples of each nematocyst type were photographed and later measured using CellSens Dimension 1.13 computer program for length and width. Linear discrimination analysis (LDA) was used to determine whether species could be distinguished on the basis of nematocyst measurements using the lda routine in the R package MASS (Venables & Ripley, 2002).

A total of 10 individuals were examined for nematocyst diversity (Table S1). Since initial estimates indicated that nematocyst diversity varied by tentacle region, nematocyst counts were done from three tentacle regions for each individual: proximal (near the base of the tentacle), medial (in the middle of the tentacle), and distal (at the end of the tentacle). For each region, the first 200 nematocysts were photographed and categorized according to nematocyst type. Only lone nematocysts were enumerated, with any nematocysts still adhering to epithelial tissue ignored, since smaller nematocysts (e.g., a-isorhizas) could be obscured. In order to examine any differences in nematocyst diversity between different tentacle regions (distal, medial, proximal), a mosaic plot showing the relative proportions of nematocyst types in the various regions was made using the R package vcd version 1.4-3 (Meyer, Zeileis & Hornik, 2016). In order to visualize differences in proportions of nematocyst types (four types, three regions) between the two species we conducted non-metric multidimensional scaling of the Euclidean distance matrix using the isoMDS routine in the R package MASS (Venables & Ripley, 2002).

Results

Sequence data characteristics and phylogenetic inference

The COI dataset consisted of 73 sequences, 59 of which are new. All sequences were 616 bp in length. The 16S data set was made up of 67 sequences, including 60 new sequences and 7 published sequences. New complete sequences varied in length from 598 base pairs (bp) for Chrysaora lactea to 608 bp (Chrysaora chinensis). The MAAFT-aligned data set (included published sequences) was 628 bp, but the dataset was truncated to 582 bp (95.7%) after treatment with GBlocks. The 28S dataset included 35 sequences, including 24 new sequences and 11 published sequences. New sequences ranged in size from 998 (Chrysaora chinensis) to 1,018 bp (Chrysaora africana). The MAFFT alignment (which included published sequences) was 1,027 bp, but the final data set was 1,015 bp (98.8%) after removal of regions via GBlocks.

All phylogenetic analyses (COI, 16S, 28S, and combined) revealed similar terminal clades, but they differed in the resolution of relationships among them. The combined analysis provided the best resolution (smallest proportion of polytomous nodes) and highest support values for evolutionary relationships (Figs. 4–7). In all analyses, Chrysaora is revealed as paraphyletic with respect to species of Sanderia, Pelagia, and Mawia. In the combined analyses, Mawia benovici is most closely related to S. malayensis (Bayesian support 100/ML support 100), with these two species forming a close relationship with Chrysaora africana and Chrysaora pacifica in the combined (88/67) and 28S trees (100/61) (Figs. 6 and 7). Except for the COI tree, P. noctiluca formed a close relationship with a clade of Pacific jellies (Chrysaora achlyos, Chrysaora colorata, Chrysaora fuscescens, and Chrysaora melanaster) with high support values (combined: 100/99; 16S: 100/92; 28S: 82/58) (Figs. 5–7). For the combined analyses (100/100) and 28S (100/100), a highly supported clade was composed of Atlantic species, including Chrysaora quinquecirrha, Chrysaora lactea, Chrysaora plocamia, Chrysaora fulgida, Chrysaora hysoscella, Chrysaora chesapeakei [see Discussion], and the Caribbean Chrysaora, while this clade was less supported for COI (100/61) and 16S (75/60) (Figs. 4–7). Chrysaora fulgida (NAM), Chrysaora plocamia (PMA), and Chrysaora hysoscella (IRE) formed a closely related group in all analyses with high support values (combined: 100/100; 28S: 100/99; COI: 100/94; 16S: 100/83). For sequences taken from Piraino et al. (2014) only, nuclear 28S sequences for Mawia benovici from the Mediterranean (ADR) occurred in the distantly related clade for P. noctiluca from the Atlantic (OVA), and a P. noctiluca from the Mediterranean (TYR) occurred in the distantly related clade for Mawia benovici from the Mediterranean (ADR) (Fig. 6).

Figure 4 Pelagiidae COI Phylogeny.

Bayesian inference (BI) COI tree reconstructed from CLUSTAL alignment using Mr. Bayes v3.2.4 and applying the GTR+I+G model of sequence evolution. Numbers adjacent to branches show bootstrap support if ≥0.70 (presented as a percentage), followed by bootstrap support from maximum likelihood (ML) analysis if ≥50%. ML phylogeny was reconstructed using PhyML v3.0 (Guindon et al., 2010) applying the TPM2uf+I+G model of sequence evolution (-lnl 5451.81154) as determined by jMODELTEST v2.1.7 (Darriba et al., 2012). Abbreviations refer to Tables 1 and 2. Specific identification to the right of the tree indicates final species designations. Clades colored in gray were originally identified as Chrysaora quinquecirrha. Norfolk (VA) individuals NF1–NF3 were identified as white Chesapeake Bay color morph and individuals NF4–NF5 as red-striped Chesapeake Bay color morph (Figs. 3D and 3E).

Figure 5 Pelagiidae 16S Phylogeny.

Bayesian inference (BI) 16S tree reconstructed from MAFFT alignment using Mr. Bayes v3.2.4 and applying the GTR+I+G model of sequence evolution. Numbers adjacent to branches show bootstrap support if ≥0.70 (presented as a percentage), followed by bootstrap support from maximum likelihood (ML) analysis if ≥50%. ML phylogeny was reconstructed using PhyML v3.0 (Guindon et al., 2010) applying the TIM2+I+G model of sequence evolution (-lnl 3641.97519) as determined by jMODELTEST v2.1.7 (Darriba et al., 2012). Gray arrows indicate nodes that are alternated in the ML tree. Abbreviations refer to Tables 1 and 2. Specific identification to the right of the tree indicates final species designations. Clades colored in gray were originally identified as Chrysaora quinquecirrha s.l. Norfolk (VA) individuals NF1–NF3 were identified as white morph and individuals NF4–NF5 as red-striped bell morphs (Figs. 3D and 3E).

Figure 6 Pelagiidae 28S Phylogeny.

Bayesian inference (BI) 28S tree reconstructed from MAFFT alignment using Mr. Bayes v3.2.4 and applying the GTR+I+G model of sequence evolution. Numbers adjacent to branches show bootstrap support if ≥0.70 (presented as a percentage), followed by bootstrap support from maximum likelihood (ML) analysis if ≥50%. ML phylogeny was reconstructed using PhyML v3.0 (Guindon et al., 2010) applying the TrNef+I+G model of sequence evolution (−lnl 3817.02691) as determined by jMODELTEST v2.1.7 (Darriba et al., 2012). Specific identification to the right of the tree indicates final species designations. Clades colored in gray were originally identified as Chrysaora quinquecirrha.

Figure 7 Pelagiidae combined phylogeny.

Bayesian inference (BI) tree of the combined dataset applying the GTR+I+G model of sequence evolution. Numbers adjacent to branches show bootstrap support if ≥0.70 (presented as a percentage), followed by bootstrap support from maximum likelihood (ML) analysis if ≥50%. ML phylogeny was reconstructed using PhyML v3.0 (Guindon et al., 2010) applying the GTR+I+G model of sequence evolution (−lnl 11924.23655) as determined by jMODELTEST v2.1.7 (Darriba et al., 2012). Specific identification to the right of the tree indicates final species designations. Clades colored in gray were originally identified as Chrysaora quinquecirrha.

At the species level, our analyses highlighted multiple species boundaries, and showed the samples identified as Chrysaora quinquecirrha to be polyphyletic. In all analyses, Chrysaora quinquecirrha sequences fell into two distinct, highly diverged clades (Figs. 4–7; Tables S3–S5), with one clade (Chrysaora chesapeakei—see “Discussion” and “Systematics”) made up of animals from U.S. Atlantic estuaries and the Gulf of Mexico animals and another (Chrysaora quinquecirrha—see “Discussion” and “Systematics”) made up of U.S. coastal Atlantic animals. Caribbean Chrysaora (Jamaica and Panama) formed a clade closely related to Chrysaora chesapeakei in all analyses (Figs. 4–7). Aquarium animals previously identified as Chrysaora melanaster (AQA) were genetically distinct from Chrysaora melanaster collected from the Bering Sea (BER) in all analyses where both were included (Figs. 4–6) and formed a clade with Chrysaora pacifica collected from South Korea (KOR) and Japan (KYO) for COI and/or 16S. While aquarium collected Chrysaora chinensis formed a well-supported clade with field collected Chrysaora chinensis (MAL), analyses differed in where this species fell out in the trees (Figs. 4–7). The unknown pelagiid collected from the Western African coast (SEN) was nearly identical to the newly described Mawia benovici from the Mediterranean for COI (0.0–0.3% difference) and 28S (0.0–0.2% difference) (Figs. 4 and 6).

Macromorphological and nematocyst analyses

A total of 120 medusae (57 field collected and 63 museum specimens) (Table S1) previously identified as Chrysaora quinquecirrha s.l. were examined for 19 quantitative and qualitative macromorphological characters taken from Gershwin & Collins (2002) and one new to this study (maximum oral arm length) (Table 3). Overall, three macromorphological characters differed significantly: tentacle number, lappet number, and maximum oral arm length vs. bell diameter (Table 3). Animals collected from the estuarine Atlantic and all Gulf of Mexico sites (Table S1) had an average of [3.07 ± 0.07] 95% CI tentacles per octant, excluding two aberrant individuals (6 and 4.625—see “Discussion”) (Fig. 8A; Table 3). In all instances when there were more than three tentacles per octant (excluding aberrant individuals above), the additional tentacle(s) occurred between the secondary tentacles and the rhopalia (i.e., 3-2-1-2-3 octant tentacle orientation) and were typically undeveloped, being of similar size to nearby lappets. Animals collected from coastal regions along the U.S. Atlantic (Table S1) had an average of [5.28 ± 0.48] 95% CI tentacles per octant (Fig. 8A; Table 3). Animals collected from the estuarine Atlantic and all Gulf of Mexico sites (Table S1) had oral arms that were on average 3.00 ± 0.39 (95% CI) times as long as the bell diameter (Fig. 8B; Table 3). Animals collected from coastal regions of the U.S. Atlantic (Table S1) had oral arms that were on average [1.24 ± 0.27] 95% CI times as long as bell diameter (Fig. 8B; Table 3). Of the animals that were examined morphologically, a total of 38 individuals were also sequenced for 16S to see which Chrysaora clade they fell into (K2P sequence divergence <1.5%). Medusae examined morphologically that fell into the Chrysaora chesapeakei phylogenetic clade had an average of 2.99 ± 0.03 tentacles per octant and oral arms that were [2.80 ± 0.78] 95% CI times as long as bell diameter on average, while all those that fell in the Chrysaora quinquecirrha clade had an average of 5.63 ± 0.78 tentacles per octant and oral arms that were on average 0.93 ± 0.18 (95% CI) times as long as bell diameter on average (Figs. 8A and 8B).

Figure 8 Morphological evidence separating Chrysaora quinquecirrha and Chrysaora chesapeakei.

(A) Tentacle counts. Graph represents tentacles per octant against bell diameter (mm) for field collected and museum specimens. Squares represent animals taken from estuarine Atlantic and Gulf of Mexico regions (Chrysaora chesapeakei), while circles represent animals taken from coastal Atlantic regions (Chrysaora quinquecirrha). All animals with 16S sequences matching the Chrysaora chesapeakei clade appear in red, while those whose sequences matched the Chrysaora quinquecirrha clade appear in blue. (B) Maximum oral arm measurements. Graph represents maximum oral arm length against bell diameter (mm) for field-collected and museum specimens. Squares represent animals taken from U.S. Atlantic estuaries and the Gulf of Mexico (Chrysaora chesapeakei), while circles represent animals taken from coastal Atlantic regions (Chrysaora quinquecirrha). Only animals with fully intact and extended oral arms were included. All animals with 16S sequences matching the Chrysaora chesapeakei clade appear in red, while those whose sequences matched the Chrysaora quinquecirrha clade appear in blue. (C) Average size measurements for holotrichous A-isrohiza nematocysts (length vs. width), based on 10 nematocysts per. Error bars represent 95% CI (2*standard error). Squares represent nematocysts from estuarine Atlantic and Gulf of Mexico medusae (Chrysaora chesapeakei), while circles represent nematocysts from coastal Atlantic medusae (Chrysaora quinquecirrha). Photograph of an average sized A-isorhiza from Chrysaora quinquecirrha appears on the left and a photograph of an average size A-isorhiza from Chrysaora chesapeakei appears on the right. Scale bars = 10 μm. Photographs have been resized so that all error bars are the same size on the page to allow size comparisons. All animals with 16S sequences matching the Chrysaora chesapeakei clade appear in red, while those whose sequences matched the Chrysaora quinquecirrha clade appear in blue. Triangles represent average values from Papenfuss (1936) for morphs identified as Dactylometra quinquecirrha (gray) or Dactylometra quinquecirrha var. chesapeakei (white).

We also studied the cnidome of medusae identified as Chrysaora quinquecirrha, examining the measurements of individual nematocyst types (Fig. 8C; Fig. S1), as well as the representation of each type overall. Nematocyst measurements indicated significant grouping for holotrichous A-isorhizas, but not for other types. A-isorhiza measurements (length vs. width) showed two distinct groups, with one group containing only animals from U.S. Atlantic estuaries and the Gulf of Mexico and the other containing coastal Atlantic animals (Fig. 8C). All sequenced animals in the smaller group (coastal Atlantic) were genetically similar to Chrysaora quinquecirrha for 16S, while all jellyfish from the larger group (estuarine Atlantic and Gulf of Mexico) that were sequenced for 16S were genetically similar to Chrysaora chesapeakei (Fig. 8C). For animals identified as Chrysaora chesapeakei (based on habitat, macromorphology, and/or genetics), LDA analysis indicated that individual A-isorhiza measurements correctly identified species 97.8% of the time (2.2% of the time, they were incorrectly identified at Chrysaora quinquecirrha), while they were correctly identified 100% of the time using the mean of 10 nematocyst measurements. For animals previously identified as Chrysaora quinquecirrha (based on habitat, macromorphology, and/or genetics), LDA correctly identified them 100% of the time, whether one or 10 nematocysts were used. Figure S2 (A–C) shows measurement graphs for a-isorhiza, O-isorhiza, and heterotrichous microbasic rhopaloids, all of which indicate no significant groupings of measurements.

Nematocysts from proximal, medial, and distal regions were typed and counted (200 total) for 10 individuals originally identified as Chrysaora quinquecirrha, chosen based on their previous molecular and macromorphological groupings (five from each group). All in all, heterotrichous microbasic rhopaloids were most frequent ([62.1 ± 9.8%] 95% CI), followed by O-isorhizas ([13.4 ± 5.0%] 95% CI), a-isorhizas ([12.4 ± 2.8%] 95% CI) and A-isorhizas ([12.2 ± 3.7%] 95% CI). As pilot studies indicated, nematocyst type proportions were different for different tentacles regions. While A-isorhizas and a-isorhizas were consistent over the entire tentacle, O-isorhizas were overrepresented in proximal regions and heterotrichous microbasic rhopaloids were overrepresented in the medial and distal regions (Fig. S3A). Individuals varied considerably in proportions of nematocyst types (Fig. S3B). Individuals collected from coastal Atlantic regions (circles) were generally clustered, including those genetically similar to Chrysaora quinquecirrha, while those from estuarine Atlantic and Gulf of Mexico regions (squares) were much more dispersed, as were those genetically similar to Chrysaora chesapeakei (Fig. S3B). LDA was moderately effective in distinguishing species using overall nematocyst proportions (four of five Chrysaora quinquecirrha and three of five Chrysaora chesapeakei correctly classified) and this was almost entirely due to different proportions of A-isorhiza nematocysts. A-isorhiza proportions were significantly different (t = 3.623, p = 0.0068), with Chrysaora chesapeakei individuals averaging 16.5 ± 3.4% for A-isorhiza and Chrysaora quinquecirrha cnidomes averaging 7.8 ± 3.4%.

Discussion

Genus-level systematic inference

Our most robust phylogenetic hypothesis for Pelagiidae (Fig. 7), based on the combined data set, directly contradicts current generic definitions, as well as earlier morphological-based phylogenies of the Pelagiidae. Both Gershwin & Collins (2002) and Morandini & Marques (2010) considered Chrysaora to be reciprocally monophyletic with respect to both Sanderia and Pelagia, with Sanderia in a basal position (Figs. 9A and 9B). In contrast, our analyses indicate that Chrysaora is paraphyletic with respect to Pelagia, Sanderia, and the newly erected Mawia (Figs. 4–7 and 9E). Mediterranean Mawia benovici is not in the combined analysis, but our Senegal pelagiid (SEN) can be treated as Mawia benovici, based on COI (Fig. 4) and 28S (Fig. 6) phylogenies (see below). Paraphyly of Chrysaora is not supported in morphological or genetic analyses in Avian et al. (2016) (Figs. 9C and 9D), but this is almost certainly a result of incomplete taxon sampling. For example, their analysis based on combined morphological and genetic data only included Chrysaora hysoscella (Mediterranean), while the 28S dataset included a subset of sequences published at the time, (Chrysaora hysoscella, Chrysaora lactea, and Chrysaora c.f. chesapeakei [see below]), all of which occur in a single clade in our analysis (Figs. 7 and 9E). Including fewer published sequences gave the appearance of Chrysaora monophyly, which may have biased the establishment of Mawia. For instance, throughout Avian et al. (2016), Chrysaora is often used as a singular entity (i.e., monophyletic), such as an entire section that examines characters at the “genus level”. This more readily allows for the conclusion of a novel genus Mawia, as it sidesteps the difficult taxonomic questions raised by the paraphyly of Chrysaora. That notwithstanding, in agreement with both Piraino et al. (2014) and Avian et al. (2016), our analyses show Mawia benovici to be a close relative of S. malayensis (Figs. 4–7). Given the stark morphological differences between Sanderia and Mawia (Piraino et al., 2014; Avian et al., 2016), this relationship is more than a bit surprising.

Figure 9 Pelagiidae evolution.

Cladograms showing genus-level relationships within the Pelagiidae family. Colors represent individual genera as shown. (A) Gershwin & Collins (2002); (B) Morandini & Marques (2010); (C) Avian et al. (2016): DNA analysis based on nuclear 28S; (D) Avian et al. (2016): morphological analyses only; (E) This study: Combined DNA analysis using sequence data from COI, 16S and 28S. *In Avian et al. (2016), this sequence is marked as Chrysaora sp. AY920779. This sequence is included in our analysis and is part of the clade that we call Chrysaora c.f. chesapeakei. ^We include the 28S phylogeny from Avian et al. (2016) because it has more species than their combined analysis but their generic conclusions are identical. Note that all previous hypotheses include a monophyletic Chrysaora.

Our working hypothesis for the relationships within Pelagiidae (Figs. 7 and 9), especially the paraphyletic Chrysaora, raises serious systematic questions for the genus level. To accept the validity of Mawia, as well as previously established Pelagia and Sanderia, each of which can be easily distinguished morphologically from those currently classified as Chrysaora, additional genera would have to be erected within Pelagiidae in order to maintain monophyly of these generic groupings. An initial matter would be to which clade should the genus Chrysaora should be limited. Because the type species of Chrysaora is Chrysaora hysoscella, the genus would best be limited to the clade containing Chrysaora hysoscella, Chrysaora fulgida, Chrysaora lactea, Chrysaora plocamia, Chrysaora quinquecirrha, and Chrysaora chesapeakei (see below). This then would leave three other lineages in need of new genera: (1) Chrysaora africana plus Chrysaora melanaster; (2) Chrysaora chinensis; and (3) Chrysaora achlyos, Chrysaora colorata, and Chrysaora fuscescens. The latter grouping (Chrysaora achlyos, Chrysaora colorata, and Chrysaora fuscescens) has a close relationship to P. noctiluca (except for COI) and there is genetic support for generic designation. Unfortunately, none of the morphological characters employed in this study clearly diagnose this clade or other Chrysaora lineages, as has been the case in other studies seeking to reconcile jellyfish taxonomy based on morphology and molecular data. (Dawson & Martin, 2001; Dawson, 2003; Bayha & Dawson, 2010). Future study will benefit from more detailed morphological analyses to identify additional characters that could then be mapped onto molecular phylogenies (e.g., Fig. 7), as well as greater taxonomic sampling (e.g., two additional Chrysaora species accepted and two declared nomen dubium in Morandini & Marques (2010), more geographic samples of Pelagia and Sanderia). Both would allow for better resolution to define genera and better explain their evolutionary relationships.

Interspecific evolutionary relationships and geographic patterns

While our molecular phylogenies bear almost no resemblance to the morphology-based phylogenies within the currently defined genus Chrysaora (Gershwin & Collins, 2002; Morandini & Marques, 2010) (Fig. 9), there are some relationships that occur in all phylogenies. All phylogenies agree on a close relationship between Chrysaora achlyos and Chrysaora colorata (Figs. 9A, 9B and 9E). Our phylogeny is in general agreement with Morandini & Marques (2010) in delineating their basal “Pacific” group (Chrysaora achlyos, Chrysaora colorata, Chrysaora fuscescens, Chrysaora melanaster, and Chrysaora plocamia), except that our Chrysaora plocamia samples came from the Atlantic and occur in an “Atlantic” group (Table 1; Fig. 1). Morandini & Marques (2010) reasoned that this basal group may have provided ancient species that then invaded the Atlantic, splitting into various Atlantic groups. Our combined phylogeny (Fig. 7) is in general agreement, with Pacific Chrysaora species generally occupying a more basal position in the tree compared to the Atlantic species. Major disagreements with Morandini & Marques (2010) include the placement of Chrysaora chinensis and Chrysaora pacifica (both Pacific jellies) as closely related to Chrysaora quinquecirrha and Chrysaora lactea, with the Chrysaora pacifica placement also a disagreement with Gershwin & Collins (2002). Likewise, the very close relationship among Chrysaora fulgida, Chrysaora hysoscella, and Chrysaora plocamia was not found in any of the morphological phylogenies (Fig. 9), though Chrysaora hysoscella and Chrysaora plocamia were closely related in Gershwin & Collins (2002).

One item of note here is our use of aquarium samples, which may be problematic where they are not confirmed with field-collected specimens. Aquarium collected specimens of Chrysaora pacifica (originally Chrysaora melanaster—see below) and Chrysaora chinensis are genetically confirmed, based on published sequences from field-collected specimens of known geographical origin (Figs. 4 and 5). In addition, our aquarium-collected Chrysaora fuscescens is identical to published 16S sequence of field-collected animals from Vancouver Island, Canada (NCBI JX393256). However, Chrysaora colorata, Chrysaora achlyos, and S. malayensis are represented only by aquarium specimens and, therefore, conclusions based on these sequences should be made with care, given questions surrounding geographic provenance and any unnatural interbreeding that might occur in an aquarium system. Future studies incorporating field-collected specimens are necessary for confirming or refuting relationships shown here.

Species-level systematic inference

Chrysaora quinquecirrha and Chrysaora chesapeakei

The most striking conclusion revealed from this study is that Chrysaora quinquecirrha, one of the most studied and well-known U.S. Atlantic jellyfish, is made up of two distinct species, putting to rest taxonomic disagreements going back more than 100 years. This finding is supported by genetic (Figs. 4–7), macromorphological (Figs. 8A and 8B), and cnidome (Fig. 8C) data. Chrysaora quinquecirrha occurred in two well-differentiated monophyletic groups, one containing all animals from estuarine Atlantic (RI, NJ, RB, NF, PAM, GA) and Gulf of Mexico (AL) regions and the other containing animals from coastal Atlantic regions (MA, CHP, and OSC) (Figs. 4–7). Average (COI: 13.1%; 16S: 9.0%; 28S: 2.5%) and minimum (COI: 12.1%; 16S: 8.4%; 28S: 2.4%; Tables S3–S5) sequence divergences are well above what has been seen as delineating species in Aurelia (Dawson & Jacobs, 2001; Dawson, Gupta & England, 2005), Cassiopea (Holland et al., 2004), Cyanea (Dawson, 2005), and Drymonema (Bayha & Dawson, 2010). More convincing is the fact that Chrysaora fulgida from Namibia (NAM), Chrysaora plocamia from Argentina (ARG), and Chrysaora hysoscella from Ireland (IRE) occur between these two species in all phylogenies (Figs. 4–7). Additionally, animals representing these genetic clades (estuarine U.S. Atlantic/Gulf of Mexico and coastal Atlantic) were consistently differentiable based on tentacle number (Fig. 8A), oral arm length (Fig. 8B), and holotrichous A-isorhiza measurements (Figs. 8C and 9). Two individuals (USNM 33457a and USNM 56703b) did not fit the typical pattern for tentacle number (Fig. 8A). However, both exhibited anomalous tentacle morphologies (multiple tentacles emerging from within lappets instead of between lappets) and had typical patterns for holotrichous A-isorhiza measurements (USNM 33457a: 27.59 × 20.98 μm; USNM 56703b: 27.04 × 21.75 μm; Fig. 8C) and/or oral arm length (USNM 33457a: 4.54 times bell diameter; USNM 56703b: sample too degraded; Fig. 8B).

It appears that Bigelow (1880) was correct that Chesapeake Bay Chrysaora that matured at 24 tentacles represented a distinct taxon from Dactylometra quinquecirrha. Our data refute the hypothesis that these individuals represent a growth stage toward the five-tentacled Chrysaora quinquecirrha described from the coast (Mayer, 1910; Calder, 1972). However, an important point is that it has been claimed that individuals only reach the “five-tentacled” stage after 130 mm (Agassiz & Mayer, 1898; Mayer, 1910), when small tentacles emerge between the secondary tentacles and the rhopalia (Mayer, 1910 Plate 64), termed Stage 5 in Calder (1972). In our data set, only a single individual larger than 130 mm was encountered and collected from the estuarine Atlantic or Gulf of Mexico (Dauphin Island, AL) and it had exactly three tentacles per octant (Fig. 8A). However, it is possible that within the estuarine Atlantic and Gulf of Mexico, these Chrysaora may develop small tertiary tentacles at very large sizes, though they likely never develop fully, as was observed in some animals examined here. Furthermore, in one case, Calder (1972) may have collected Chrysaora from an area (Broadkill River, DE) that experiences both species, albeit at different times of the day, seemingly supporting the hypothesis of development stages. The mouth of the Broadkill River experiences tidal inflows capable of pulling coastal Chrysaora into the inlet during high tide and outflows capable of pulling estuarine Chrysaora from the intercoastal waterway during low tide (K. M. Bayha, 1994–2004, personal observation). In any case, the growth of small tertiary tentacles in large estuarine Atlantic and Gulf of Mexico Chrysaora, along with the dependence on a single morphological character (tentacle number), likely led to the historical taxonomic uncertainties we are clarifying here.

Several lines of evidence support the U.S. Atlantic coastal Chrysaora group retaining the species name Chrysaora quinquecirrha and the estuarine Atlantic/Gulf of Mexico group requiring a different name. First, Pelagia quinquecirrha (=C. quinquecirrha) (Desor, 1848) was described from a coastal zone region (Nantucket Harbor, MA) as having 40 tentacles and our coastal Atlantic animals were characterized by possessing 40 or more tentacles. Furthermore, one of our sampling sites and a museum specimen were from coastal waters (Buzzard’s Bay, MA) near the Chrysaora quinquecirrha type locality. Assigning a species name to the U.S. Atlantic estuaries/Gulf of Mexico group is more problematic, owing to inconsistencies in Papenfuss (1936). Papenfuss (1936) compared two color morphs found within the Chesapeake Bay, a small, white morph (e.g., Fig. 3D) and a larger red-striped morph (e.g., Fig. 3E), which the author assumed to be Dactylometra (=Chrysaora) quinquecirrha. Papenfuss (1936) assigned the white morph to the new subspecies Dactylometra quinquecirrha var. chesapeakei, based on very small differences in holotrichous a-isorhiza measurements, though without statistical support. However, for our Norfolk (VA) samples, white (NF1–NF3) and red-striped (NF4–NF5) morphs occurred in the same genetic clades for 16S and COI (Figs. 4 and 5) and we found no overall pattern of differentiation in our holotrichous a-isorhiza measurements (Fig. S2A). Furthermore, for holotrichous A-isorhiza measurements, both morphs from Papenfuss (1936) are consistent with our U.S. Atlantic estuary/Gulf of Mexico group (Fig. 8C). In summary, evidence from nematocyst measurements (Fig. 8C), locality (Chesapeake Bay), and phylogenetic data (Figs. 4 and 5) support the U.S. Atlantic estuarine/Gulf of Mexico group and both morphs from Papenfuss (1936) as representing the same species. Even though Papenfuss (1936) may have been mistaken in describing Dactylometra quinquecirrha var. chesapeakei, that name is taxonomically available based on Article 45.6.4 of the International Code of Zoological Nomenclature (ICZN, 1999). As such, all animals from the U.S. Atlantic estuary/Gulf of Mexico lineage should be assigned to the elevated species name Chrysaora chesapeakei (Papenfuss, 1936). The placement of Gulf of Mexico medusae in Chrysaora chesapeakei differs from Morandini & Marques (2010), who placed them in the species Chrysaora lactea, based on similarities in octant tentacle orientation (2-3-1-3-2). However, our genetic data clearly separate these animals from the distantly related Chrysaora lactea (Figs. 4–7) and the number of tentacles (approximately three) and lack of tertiary tentacles in the Gulf of Mexico animals observed here and in Morandini & Marques (2010) (USNM 49733 and USNM 53826) make accurate determination of tertiary tentacle orientation problematic.

In addition to their taxonomic value, it is possible that some of the morphological characters that delineate Chrysaora quinquecirrha and Chrysaora chesapeakei may be related to adaptations to different predominant prey items, especially for feeding on the ctenophore Mnemiopsis leidyi. In general, Mnemiopsis leidyi, which is a major prey item for Chrysaora (Feigenbaum & Kelly, 1984), exhibits an inshore, estuarine preference and a seasonal spread from estuarine to coastal waters (Costello et al., 2012; Beulieu et al., 2013). As such, Mnemiopsis leidyi may be a more frequent prey item for estuarine Atlantic Chrysaora than for coastal animals. Larger oral arms, as exhibited in Chrysaora chesapeakei (Fig. 8B), have been argued to be an adaptation for scyphozoans that feed on gelatinous prey (Bayha & Dawson, 2010). In addition, the larger and more numerous A-isorhiza nematocysts found in estuarine Chrysaora might be better suited to efficiently attaching to and feeding on very soft-bodied organisms such as Mnemiopsis leidyi. Since different nematocyst types are assumed to have different functions based on morphological and discharge characteristics (Rifkin & Endean, 1983; Purcell, 1984; Heeger & Möller, 1987; Purcell & Mills, 1988; Colin & Costello, 2007), it has been proposed that nematocyst diversity within an organism can be correlated to dietary preferences, at least in a coarse sense (Purcell, 1984; Purcell & Mills, 1988; Carrette, Alderslade & Seymour, 2002). In particular, isorhiza nematocysts, which typically serve to entangle hard prey or penetrate soft tissue (Purcell & Mills, 1988; Colin & Costello, 2007), are likely important for feeding on gelatinous prey, since they are the only types found in some jelly-feeding medusae, such as hydrozoan narcomedusae (Purcell & Mills, 1988) and the scyphozoan Drymonema larsoni (KM Bayha, personal observation). A-isorhizas are about twice as numerous in Chrysaora chesapeakei as in Chrysaora quinquecirrha (16.5 ± 3.4% vs. 7.8 ± 3.4%) and are significantly larger (Fig. 8C) in Chrysaora chesapeakei. It is possible that the more numerous A-isorhizas, possessing longer tubules, could penetrate farther into the extremely soft-bodied Mnemiopsis leidyi, resulting in greater capture efficiency.

Chrysaora in the Caribbean

Chrysaora medusae collected from the Caribbean Sea are genetically very similar to Chrysaora chesapeakei. Chrysaora in the Caribbean have historically been included in the species C. lactea (Mayer, 1910; Morandini & Marques, 2010), C. quinquecirrha (Perry & Larson, 2004), or Chrysaora sp. (Persad et al., 2003). Our Caribbean samples, limited only to Jamaica and the Bocas del Toro region of Panama, appear to be two lineages (both found in JAM) slightly diverged from each other (4.4–5.1% for COI) and from Chrysaora chesapeakei (6.2–7.7% for COI) from the U.S. east coast estuaries and the Gulf of Mexico. These animals cannot be assigned to Chrysaora lactea (type locality = Rio de Janiero, Brazil), as was previously done by Mayer (1910) and Morandini & Marques (2010), since these animals are distantly related to Chrysaora lactea for most genetic regions examined (Figs. 4–7). At present, it is unclear if the Caribbean forms represent distinct or incipient species and further study of them from across the region is necessary. For the time being, we advocate referring to Caribbean animals as Chrysaora c.f. chesapeakei ahead of a formal systematic redescription based on genetic and careful morphological examination.

Chrysaora melanaster and Chrysaora pacifica

Our phylogenetic data confirm the morphological conclusions in Morandini & Marques (2010) that Japanese Chrysaora historically identified as Chrysaora melanaster, and labeled as such in public aquaria worldwide for decades, are actually the distinct species Chrysaora pacifica. Kramp (1961) synonymized the Pacific Chrysaora species C. melanaster (Brandt, 1835) and the Japanese jellyfish C. pacifica (Goette, 1886) to Chrysaora melanaster. This identification convention made it into jellyfish identification books (Wrobel & Mills, 1998) and subsequently Japanese Chrysaora labeled as Chrysaora melanaster became a mainstay in early jellyfish exhibits, such as the Monterey Bay Aquarium (MBA), and then in aquaria throughout the world (W Patry, personal communication). Morandini & Marques (2010) separated Chrysaora melanaster and Chrysaora pacifica on morphological grounds (tentacle and lappet number) and deemed all aquarium specimens of Japanese origin to be Chrysaora pacifica. Our data (Figs. 4 and 5) confirm this, as aquarium-collected jellyfish previously labeled Chrysaora melanaster (MBA) are distantly related to wild-caught Chrysaora melanaster (BER) from its type locality (Bering Sea), but are nearly genetically similar (sequence divergence: COI: 0.5%; 16S: 0.6%) to wild-caught Chrysaora collected from the Eastern Korean coast (KOR), where this jellyfish was recently redescribed as C. pacifica (Lee et al., 2016) and Kyoto, Japan (KYO), both near the type locality of Nagasaki, Japan (Goette, 1886).

Chrysaora africana/fulgida

Our phylogenies support the resurrection of Chrysaora species along the southwestern coast of Africa. Three species of Chrysaora were previously identified from the southwestern coast of Africa: Chrysaora hysoscella (Kramp, 1955), C. fulgida (Reynaud, 1830) and C. africana (Vanhöffen, 1902). However, Kramp (1961) deemed Chrysaora africana a variant of Chrysaora fulgida, and Morandini & Marques (2010) placed all of these sightings within the species Chrysaora fulgida. All phylogenies indicate two distantly related species of Chrysaora from Namibian waters (Figs. 4–7), with those appearing superficially similar to Chrysaora fulgida (brown striped) or to Chrysaora africana (red tentacles) placed provisionally into these species. These designations are consistent with upcoming redescriptions of Chrysaora fulgida and Chrysaora africana of S. Neethling, 2014, unpublished data based on morphological and genetic analyses. Chrysaora has increased over recent years in this area, with concomitant ecological perturbations (Lynam et al., 2006; Flynn et al., 2012; Roux et al., 2013), underscoring the importance of correct species identification.

Mawia benovici

In addition to revealing higher level phylogenetic relationships, our data add to our knowledge regarding the distribution of Mawia benovici, indicating a possible source region for the introduced species. Piraino et al. (2014) hypothesized that Mawia benovici (then Pelagia benovici), likely arrived into the Adriatic Sea via ballast water. Our data indicate that two small pelagiid jellyfishes collected from the beach near Dakar, Senegal are Mawia benovici based on COI and 28S phylogenies (Figs. 4 and 6) (there are no published 16S sequences for Mawia benovici). While this is not definitive evidence that Mediterranean Mawia benovici populations originated from the western coast of Africa, it raises the possibility. While many West African species have arrived in the Mediterranean through the Strait of Gibraltar or occasionally inhabit the Western Mediterranean (Gofas & Zenetos, 2003; Antit, Gofas & Azzouna, 2010), there are examples of animals introduced via shipping or fishing practices from West Africa to the Mediterranean (Ben Souissi et al., 2004; Antit, Gofas & Azzouna, 2010; Luque et al., 2012; Zenetos et al., 2012). If Mawia benovici did originate from the western coast of Africa, it is more likely that it was a result of shipping or fishing practices, since there are no records of Mawia benovici between Gibraltar and the Adriatic Sea to our knowledge.

Systematics

Chrysaora quinquecirrha Desor, 1848

Figs. 3A, 3B, 4–9; Figs. S1 and S2.

Pelagia quinquecirrha-Desor (1848): p. 76 (original description—Nantucket Sound, MA).

Dactylometra quinquecirrha: Agassiz (1862): 126, 166 [tentacle number]. Agassiz (1865): 48, 49 [tentacle number; Naushon, MA]. Fewkes (1881): 173, Pl. VIII Fig. 14 [tentacle number, drawing]. Brooks (1882): 137 [tentacles, drawing in Mayer, 1910; southern variety outside Beaufort Inlet]. Agassiz & Mayer (1898): 1–6, Plate I [tentacles, oral arms, drawing]. Fish (1925): 128, 130 [Vineyard Sound, MA; Nonamesset, MA; Lackeys Bay, MA]. Mayer (1910): 585–588, Pl. 64A [tentacles, drawing].

Chrysaora quinquecirrha: Kramp (1961): 327–328 [description fits both Chrysaora quinquecirrha and Chrysaora chesapeakei]. Calder (1972): 40–43, Figures 5–6 [mouth of Broadkill River, DE]. Kraeuter & Setzler (1975): 69, Figures 1–2 [offshore samples, Sea Buoy]. Calder (2009): 24–28 [offshore animals collected on continental shelf possibly Chrysaora quinquecirrha].

Diagnosis: Living medusae up to 40 cm (observed 59.0–176.0 mm) (Figs. 3A and 3B); tentacles typically 40 or more; 5.28 ± 0.45 (95% CI) tentacles/octant on average (Table 3; Fig. 8A); lappets rounded typically 48 or more; 6.26 ± 0.46 lappets/octant on average; rhopalar lappets slightly larger than tentacular lappets; can be differentiated from Chrysaora chesapeakei based on (1) smaller size of holotrichous A-isorhiza nematocysts: average: 20.25 [±0.38] × 11.27 [±0.37] μm (Table 3; Fig. 8C); (2) larger tentacle number (more than five per octant); and (3) typically shorter maximum oral arm length (average: 1.24 ± 0.27 time bell diameter).

Material examined: USNM 24496 (n = 1; Buzzard’s Bay, MA), USNM 53860 (n = 1; Assateague Island, VA), USNM 53861 (n = 1; Assateague Island, VA), USNM 54511 (n = 2; Cape Henlopen, DE), USNM 56702 (n = 1; Cape Henlopen, DE), USNM 1454776–USNM 1454778, KMBCDE2, KMBCDE4 (n = 5; Cape Henlopen, DE).

Description of holotype: No holotype located, no neotype designated.

Description of specimens: Bell diameter up to approximately 40 cm (observed 59.0–176.0 mm), almost hemispherical. Exumbrella finely granulated with small, inconspicuous marks (papillae); exumbrellar color varies from entirely transparent white to white with inconspicuous radial markings. Tentacle number approximately five tentacles per octant, but can be more (average 5.28 ± 0.48) (Table 3; Fig. 8A); lappets rounded typically 48 or more (average 6.26 ± 0.46 per octant); tentacle clefts of varied depth with primary clefts deeper than secondary clefts. Radial and ring musculature not obvious. Brachial disc circular. Pillars evident. No quadralinga. Subgenital ostia rounded, approximately 1/8 of bell diameter. Oral arms v-shaped with frills emanating from tube-like structure; straight without spiral; curved, frilled edges taper toward distal end of oral arms. Oral arms short, approximately the same length as bell diameter (average 1.24 ± 0.27 times bell diameter). Oral arms typically transparent white. Four semi-circular gonads, white, pinkish, or slightly orange, well developed within pouch outlining gastric filaments. About 16 stomach pouches bounded by 16 septae. Septae bent at 45° angle distally toward the rhopalia terminating near tentacle in rhopalar lappet, resulting in tentacular pouches being somewhat larger than rhopalar pouches distally.

Cnidome (tentacle): Average dimensions (length ± 95% CI × width ± 95% CI) Holotrichous A-isorhizas: 20.15 ± 0.33 × 11.13 ± 0.24 μm;

Holotrichous a-isorhizas: 8.27 ± 0.49 × 4.22 ± 0.07 μm;

Holotrichous O-isorhizas: 21.63 ± 0.39 × 18.91 ± 0.78 μm;

Heterotrichous microbasic rhopaloids: 13.58 ± 0.19 × 8.09 ± 0.09 μm;

Type locality: Nantucket Bay, Nantucket Island, Massachusetts, East Coast of USA.

Habitat: Medusae are found in open coastal waters on the U.S. Atlantic coast.

Distribution: Western North Atlantic, east coast of the USA south of Cape Cod in coastal Atlantic waters at least as far south as Georgia/Northern Florida.

DNA sequence: Mitochondrial COI and 16S and nuclear 28S sequence data are available in NCBI GenBank under accession numbers MF141552–MF141556, MF141608, MF141613–MF141614, MF141628, MF141635, MF141642–MF141646, MF141688–MF141689, MF141697.

Phylogeny: Chrysaora quinquecirrha and Chrysaora chesapeakei sequences form reciprocally monophyletic groups for 16S, COI, 28S, and combined analyses (Figs. 4–7). Minimum sequence divergences between Chrysaora quinquecirrha and Chrysaora chesapeakei clades (COI: 12.1%, 16S: 8.4%, 28S: 2.4%) were much larger than the maximum within clades for Chrysaora quinquecirrha (COI: 0.2%, 16S: 0.1%, 28S: 0.0%) or Chrysaora chesapeakei (COI: 0.7%, 16S: 0.6%, 28S: 0.4%). Chrysaora quinquecirrha sequences did not form monophyletic groups with any other species (Figs. 4–7).

Biological data: Although the name Chrysaora quinquecirrha applies to the U.S. coastal Atlantic species, almost no ecological studies have been done on the coastal species, apart from (Kraeuter & Setzler, 1975), which found the largest Chrysaora quinquecirrha individual was found in a coastal area about 90 km offshore in full seawater (salinity >30).

Notes: Since this species retains the scientific name Chrysaora quinquecirrha, we advocate it retaining the common name “U.S. Atlantic sea nettle”, since it is also a coastal and open ocean species.

Chrysaora chesapeakei Papenfuss, 1936

Figs. 3C–3E and 4–9; Figs. S1 and S2

Dactylometra quinquecirrha: Bigelow (1880): 66 [white colored morph, Chesapeake Bay]. Brooks (1882): 137 [Chesapeake Bay—USA]. Agassiz & Mayer (1898): 48–49 [upper Narragansett Bay (RI)]. Mayer (1910): 585–588, Pl.63–64 [24 tentacle morph, white, red/brown striped morph, Tampa Bay (FL), Hampton Roads (VA), St. Mary’s (MD)]. Papenfuss (1936): 14–17, Figures 7, 11, 16, 20 [lower Chesapeake Bay; red-striped morph based on A-isorhiza measurements]. Littleford & Truitt (1937): 91 [Chesapeake Bay]. Littleford (1939): 368–381, Pls. I–III [Chesapeake Bay]. Hedgepeth (1954): 277–278 [Tampa Bay (FL), Gulf of Mexico].

Dactylometra quinquecirrha var. chesapeakei: Papenfuss (1936): 14–17, Figures 12, 21 [Chesapeake Bay; white colored morph based on A-isorhiza measurements].

Chrysaora quinquecirrha: Kramp (1961): 327–328 [parts of description covers both Chrysaora quinquecirrha and Chrysaora chesapeakei]. Rice & Powell (1970): 180–186 [Chesapeake Bay]. Burke (1976): 20, 22–28 [Mississippi Sound, Gulf of Mexico]. Calder (1971): 270–274 [Gloucester Point (VA)—Chesapeake Bay]. Calder (1972): 40–43, Figures 1–4 [Chesapeake Bay, Pamlico Sound, Gulf of Mexico]. Loeb (1972): 279–291 [Chesapeake Bay]. Loeb (1973): 144–147 [Chesapeake Bay]. Loeb & Blanquet (1973): 150–157 [Chesapeake Bay]. Calder (1974b): 326–333 [Chesapeake Bay]. Loeb (1974): 423–432 [Chesapeake Bay]. Blanquet & Wetzel (1975): 181–192 [Chesapeake Bay]. Cargo (1975): 145–154 [Chesapeake Bay]. Kraeuter & Setzler (1975): 69, Figures 1–2 [Doboy Sound (GA)]. Loeb & Gordon (1975): 37–41 [Chesapeake Bay]. Lin & Zubkoff (1976): 37–41 [Chesapeake Bay]. Calder (1977): 13–19 [Gloucester Point, MD—Chesapeake Bay]. Clifford & Cargo (1978): 58–60 [Patuxent River, MD—Chesapeake Bay]. Cargo (1979): 279–286 [Chesapeake Bay]. Cargo & Rabenold (1980): 20–26 [Patuxent River (MD)]. Hutton et al. (1986): 154–155 [Chesapeake Bay]. Cargo & King (1990): 486–491 [Chesapeake Bay]. Purcell et al. (1991): 103–111 [Choptank River, MD—Chesapeake Bay]. Nemazie, Purcell & Glibert (1993): 451–458 [Chesapeake Bay]. Purcell, White & Roman (1994): 263–278 [Chesapeake Bay]. Burnett et al. (1996): 1377–1383 [Chesapeake Bay]; Houck et al. (1996): 771–778 [St. Margaret’s, MD—Chesapeake Bay]. Olesen, Purcell & Stoecker (1996): 149–158 [Broad Creek (MD)—Chesapeake Bay]. Ford et al. (1997): 355–361 (Choptank River (MD)—Chesapeake Bay]. Kreps, Purcell & Heidelberg (1997): 441–446 [Choptank River (MD)—Chesapeake Bay]. Wright & Purcell (1997): 332–338 [Patuxent River (MD)—Chesapeake Bay]. Suchman & Sullivan (1998): 237–244 [Green Hill Pond (RI)]. Purcell, Malej & Benović (1999): 241–263 [Chesapeake Bay]. Purcell et al. (1999): 187–196 [Choptank River (MD)—Chesapeake Bay]. Bloom, Radwan & Burnett (2001): 75–90 [St. Mary’s (MD)—Chesapeake Bay]. Condon, Decker & Purcell (2001): 89–95 [Choptank River (MD)—Chesapeake Bay]. Graham (2001): 97–111 [Gulf of Mexico]. Johnson, Perry & Burke (2001): 213–221 [Gulf of Mexico]. Matanoski, Hood & Purcell (2001): 191–200 [Choptank River (MD)—Chesapeake Bay]. Segura-Puertas, Suárez-Morales & Celis (2003): 9 [Gulf of Mexico]. Ishikawa et al. (2004): 895–899 [Gibson Island (MD)—Chesapeake Bay]. Grove & Breitburg (2005): 185–198 [Patuxent River (MD)—Chesapeake Bay]. Purcell & Decker (2005): 376–385 [Chesapeake Bay]. Thuesen et al. (2005): 2475–2482 [Chesapeake Bay]. Breitburg & Fulford (2006): 776–784 [Solomon’s Island [MD]—Chesapeake Bay]. Kimmel, Roman & Zhang (2006): 131–141 [mid to upper Chesapeake Bay]. Decker et al. (2007): 99–113 [Chesapeake Bay]. Condon & Steinberg (2008): 153–168 [York River (VA)—Chesapeake Bay]. Calder (2009): 24–28 [estuarine animals]. Matanoski & Hood (2006): 595–608 [Choptank River (MD)—Chesapeake Bay]. Purcell (2007): 184, 190–192 [Chesapeake Bay]. Purcell (2009): 23–50 [Chesapeake Bay]. Duffy, Epifanio & Fuiman (1997): 123–131 [Port Aransas (TX)—Gulf of Mexico]. Bayha & Graham (2009): 217–228 [Rhode Island, New Jersey, Chesapeake Bay, Georgia, Alabama]. Sexton et al. (2010): 125–133 [Choptank River (MD)—Chesapeake Bay]. Birsa, Verity & Lee (2010): 426–430 [Skidaway River (GA), Wassow Sound (GA)]. Condon, Steinberg & Bronk (2010): 153–170 [York River (VA)—Chesapeake Bay]. Condon et al. (2011): 10225–10230 [Chesapeake Bay]. Frost et al. (2012): 247–256 [Steinhatchee River (FL)—Gulf of Mexico]. Duarte et al. (2012): 91–97 [St. Leonard’s (MD)—Chesapeake Bay]. Kimmel, Boynton & Roman (2012): 76–85 [Solomon’s Island (MD)—Chesapeake Bay]. Sexton (2012): 1–153 [Chesapeake Bay]. Brown et al. (2013): 113–125 [Chesapeake Bay]. Robinson & Graham (2013): 235–253 [Gulf of Mexico]. Breitburg & Burrell (2014): 183–200 [Patuxent River (MD)—Chesapeake Bay]. Kaneshiro-Pineiro & Kimmel (2015): 1965–1975 [Pamlico Sound (NC). Meredith, Gaynor & Bologna (2016): 6248–6266 [Barnegat Bay (NJ)]. Tay & Hood (2017): 227–242 [Choptank River (MD), Chesapeake Bay].

Diagnosis: Living medusae up to 20 cm (observed 17.0–175.0 mm; average: 63.0 mm); tentacles typically number 24 or 3 per octant (average 3.07 ± 0.07); primary tentacle central and secondary tentacles lateral (2-1-2); rarely, additional tentacles arise lateral to secondary tentacles (3-2-1-2-3) and are typically undeveloped; marginal lappets rounded and typically 32 or 4 per octant (average 4.08 ± 0.06); rhopalar lappets are typically about the same size as tentacular lappets; can be differentiated from Chrysaora quinquecirrha based on (1) larger size of holotrichous A-isorhiza nematocysts: 26.21 [±0.50] × 19.74 [±0.55] μm; (2) smaller tentacle number (∼3 tentacles per octant); and (3) larger maximum oral arm length (average: 3.00 ± 0.39 times bell diameter).

Material examined: Neotype: USNM 1454948—(Gloucester Point, MD—Chesapeake Bay). Other comparative specimens: USNM 57925 (n = 9; Orange Inlet, NC), USNM 56758 (n = 5; Charlestown Pond, RI), USNM 33456 (n = 4; Plum Point, MD), USNM 49733 (n = 1; Alligator Harbor, FL), USNM 53826 (n = 2; Timbalier Bay, LA), USNM 56703 (n = 2; Chesapeake Bay 37.23 N 76.04 W), USNM 56704 (n = 4; Chesapeake Bay 37.23 N 76.04 W), USNM 53870 (n = 3; Beaufort, NC), USNM 53828 (n = 2; Drum Point, MD), USNM 33458 (n = 3; Plum Point, MD), USNM 33457 (n = 4; Plum Point, MD), USNM 55621 (n = 6; near Chesapeake Beach, MD), USNM 53867 (n = 1; Arundel on the Bay, MD), USNM 54404 (n = 1; Chesapeake Bay 37.23 N 76.04 W), USNM 33121 (n = 6; Arundel on the Bay, MD), USNM 42155 (n = 2; Louisiana, Gulf of Mexico), USNM 54372 (n = 1; Lake Pontchartrain, LA); USNM 1454941–USNM 1454943, KMBCSC1, KMBCSC4–KMBCSC5, KMBCSC7 (n = 7; Charleston Harbor, SC), USNM 1454944–USNM 1454951, KMBGVA1, KMBGVA5, KMBGVA7, KMBGVA10 (n = 12; Gloucester Point, VA), KMBCRI1–KMBCRI14 (n = 14; Charlestown Pond, RI), KMBRDE1–KMBRDE16 (n = 16; Rehoboth Bay, DE), USNM 1454956, KMBDIA2–KMBDIA3 (n = 3; Dauphin Island, AL).

Description of neotype specimen: USNM 1454948. Bell diameter 110.4 mm, almost hemispherical. Exumbrella white/clear with granulated surface of small white marks. Eight rhopalia. No ocelli. Deep rhopalar clefts; deep sensory pits. Marginal lappets rounded, 32 total or 4 per octant made up of two rhopalar lappets and two tentacular lappets. Lappet size barely heterogeneous, with rhopalar lappets about the same width as tentacular lappets but longer. Tentacle number 24 or 3 per octant, with primary tentacle surrounded by two secondary tentacles (2-1-2), primary tentacle longer than secondary tentacles, up to 3–4 times bell diameter. Tentacles are white, slightly pinkish. Tentacle clefts of varied depth with primary clefts deeper than secondary clefts. Radial and ring musculature not obvious. Brachial disc circular. Pillars evident. No quadralinga. Subgenital ostia rounded, approximately 1/10 of bell diameter. Oral arms white, v-shaped with frills emanating from tube-like structure. Oral arms straight without spiral curved, frilled edges taper toward distal end of oral arms. Orals arms long, approximately five (4.98) times bell diameter. Four semi-circular gonads, white (a bit orange), well developed within pouch outlining gastric filaments. About 16 stomach pouches bounded by 16 septae. Septae bent at 45° angle distally toward the rhopalia terminating near tentacle in rhopalar lappet, resulting in tentacular pouches being somewhat larger than rhopalar pouches distally.

Cnidome (tentacle): Average dimensions (length ± 95% CI × width ± 95% CI) Holotrichous A-isorhizas: 25.66 ± 0.83 × 19.16 ± 0.54 μm;

Holotrichous a-isorhizas: 7.77 ± 0.20 × 4.17 ± 0.10 μm;

Holotrichous O-isorhizas: 22.02 ± 0.30 × 19.95 ± 0.24 μm;

Heterotrichous microbasic rhopaloids: 12.35 ± 0.47 × 8.55 ± 0.55 μm.

Description of other specimens: Bell diameter up to approximately 20 cm (observed 17.0–175.0 mm), almost hemispherical but flattened in small individuals. Exumbrella finely granulated with small, inconspicuous marks (papillae); exumbrellar color varies considerably, varying from all white to a completely brown or red colored bell, to a bell with radial lines of red/brown with a spot in the center of the bell. Radial lines may be relatively inconspicuous without a noticeable spot in the center. Tentacles typically number 24 or 3 per octant (average 3.07 ± 0.07), with primary tentacle surrounded by two secondary tentacles (2-1-2), primary tentacle longer than secondary tentacles, up to 3–4 times bell diameter. In some rare cases, small tentacles may occur laterally to secondary tentacle, occurring between the secondary tentacle and rhopalium. In almost all cases, this tentacle is similar in size to or smaller than the lappets surrounding it. In very rare cases (twice observed), about five or more tentacles per octant have been seen, though these medusae had aberrant tentacle patterns overall (e.g., more than one tentacle emerging from same spot, tentacles emerging below lappet). Tentacles are white, slightly pinkish. Marginal lappets rounded and typically 32 or 4 per octant (average 4.08 ± 0.06). Tentacle clefts of varied depth with primary clefts deeper than secondary clefts, which are deeper than rare tertiary clefts. Radial and ring musculature not obvious. Brachial disc circular. Pillars evident. No quadralinga. Subgenital ostia rounded, approximately 1/10 of bell diameter. Oral arms v-shaped with frills emanating from tube-like structure; straight without spiral; curved, frilled edges taper toward proximal end of oral arms. Oral arms long, approximately three times bell diameter on average (as much as 5.6 times bell diameter). Oral arms vary in color, from transparent white, to red or brown colored tubule surrounded by pinkish frilled edges. Four semi-circular gonads, white, pinkish or slightly orange, well developed within pouch outlining gastric filaments. About 16 stomach pouches bounded by 16 septae. Septae bent at 45° angle distally toward the rhopalia terminating near tentacle in rhopalar lappet, resulting in tentacular pouches being somewhat larger than rhopalar pouches distally.

Cnidome (tentacle): Average dimensions (length ± 95% CI × width ± 95% CI) Holotrichous A-isorhizas: 26.21 ± 0.50 × 19.74 ± 0.55 μm;

Holotrichous a-isorhizas: 7.88 ± 0.13 × 4.29 ± 0.07 μm;

Holotrichous O-isorhizas: 23.10 ± 0.43 × 20.75 ± 0.62 μm;

Heterotrichous microbasic rhopaloids: 12.73 ± 0.22 × 8.29 ± 0.13 μm.

Type locality: Gloucester Point (VA), Chesapeake Bay, east coast of USA.

Habitat: Medusae are found in estuarine waters on the U.S. Atlantic coast and estuarine and nearshore waters of the Gulf of Mexico.

Distribution: Western North Atlantic, east coast of the USA south of New England to the Gulf of Mexico, restricted to estuarine waters on the Atlantic coast, known to exist outside of estuaries in the Gulf of Mexico.

Notes: Since Chrysaora chesapeakei is commonly found in estuarine waters, we advocate the common name “Atlantic bay nettle” to distinguish it from the “U.S. Atlantic sea nettle” (Chrysaora quinquecirrha). The specific name chesapeakei originates from Dactylometra quinquecirrha var. chesapeakei of Papenfuss (1936). For Papenfuss (1936), it is clear that: (1) the manuscript likely compared nematocyst measurements between two color morphs of Chrysaora chesapeakei and did not include Chrysaora quinquecirrha s. str. (see Discussion; Fig. 8C); and (2) differences invoked for holotrichous a-isorhizas are in question, since the nematocysts are small (∼1.5 μm), making identifying differences difficult even with more precise, modern instruments, and the data are not accompanied by any statistics or measurement error. Regardless, based on Article 35.6.4 of the International Code of Zoological Nomenclature 4th Edition (ICZN, 1999), the specific name chesapeakei has taxonomic priority and Chrysaora chesapeakei applies to the Chesapeake Bay animals, as well as estuarine Atlantic and Gulf of Mexico animals that are genetically similar, and have similar macromorphological and cnidome characteristics (Figs. 4–9). Papenfuss (1936) did not designate a type specimen for Dactylometra (=Chrysaora) quinquecirrha var. chesapeakei. We designate the specimen USNM 1454948 as a neotype specimen so that a physical specimen, along with preserved tissue for genetic analysis, will be available to objectively define Chrysaora chesapeakei [see Article 75 of the International Code for Zoological Nomenclature (ICZN, 1999)], which will be necessary given the close genetic relationship between this species and specimens from the Caribbean (see below). Our neotype specimen originates from Gloucester Bay (VA), within the Chesapeake Bay, where Papenfuss (1936) hypothesized Dactylometra (=Chrysaora) quinquecirrha var. chesapeakei to be confined.

DNA sequence: Mitochondrial COI and 16S and nuclear 28S sequence data are available in GenBank under accession numbers MF141564–MF141587, MF141615–MF141617, MF141637–MF141639, MF141649–MF141669, MF141699–MF141718, MF167556–MF167568.

Phylogeny: Chrysaora chesapeakei and Chrysaora quinquecirrha sequences form reciprocally monophyletic groups for 16S, COI, 28S, and combined analyses (Figs. 4–7). Minimum sequence divergences between Chrysaora chesapeakei and Chrysaora quinquecirrha clades (COI: 12.1%, 16S: 8.4%, 28S: 2.5%) were much larger than the maximum within clades for Chrysaora quinquecirrha (COI: 0.3%, 16S: 0.1%, 28S: 0.0%) or Chrysaora chesapeakei (COI: 2.2%, 16S: 1.9%, 28S: 0.7%). Chrysaora chesapeakei sequences do not form monophyletic groups with any other species (Figs. 4–7).

Supplemental Information

Supplemental Information 1 Table S1: Geographic source regions of samples used for morphological analyses in this study.

Collected samples were collected by the authors while museum specimens all came from the Smithsonian National Museum of Natural History. Sites were categorized between coastal and estuarine regions based on geography, knowledge of regions and average salinity where available from data buoys, but deemed coastal if in doubt. *For collected regions, some specimens were sequenced for mitochondrial 16S to assign to a species/clade in Fig. 5.

Click here for additional data file.

Supplemental Information 2 Table S2: PCR primers employed in this study (A = PCR amplification; S = DNA Sequencing).

Click here for additional data file.

Supplemental Information 3 Table S3: Pairwise genetic distance matrix (COI) for major clades/species in the Pelagiidae.

Values below the diagonal are minimum pairwise genetic distances computed using the Kimura 2-parameter substitution model (Kimura, 1980) in MEGA 7.0.14 (Kumar, Stecher & Tamura, 2016). Values in bold represent maximum within clade divergences. Column/row numbers represent major species/clades: 1. Chrysaora achlyos, 2. C. africana, 3. C. chesapeakei, 4. Chrysaora c.f. chesapeakei, 5. C. chinensis, 6. C. colorata, 7. C. fulgida, 8. C. fuscescens, 9. C. hysoscella, 10. C. lactea, 11. C. melanaster, 12. C. pacifica, 13. C. plocamia, 14. C. quinquecirrha, 15. Chrysaora sp. 1, 16. Pelagia benovici, 17. P. noctiluca, 18. Sanderia malayensis, 19. Cyanea capillata.

Click here for additional data file.

Supplemental Information 4 Table S4: Pairwise genetic distance matrix (16S) for major clades in the Pelagiidae.

Values below the diagonal are minimum pairwise genetic distances computed using the Kimura 2-parameter substitution model (Kimura, 1980) in MEGA 7.0.14 (Kumar, Stecher & Tamura, 2016). Values in bold represent maximum within clade divergences. Column/row numbers represent major taxa/clades: 1. Chrysaora achlyos, 2. C. africana, 3. C. chesapeakei, 4. Chrysaora c.f. chesapeakei, 5. C. chinensis, 6. C. colorata, 7. C. fulgida, 8. C. fuscescens, 9. C. hysoscella, 10. C. lactea, 11. C. melanaster, 12. C. pacifica, 13. C. plocamia, 14. C. quinquecirrha, 15. Pelagia benovici, 16. P. noctiluca, 17. Sanderia malayensis, 18. Cyanea capillata.

Click here for additional data file.

Supplemental Information 5 Table S5: Pairwise genetic distance matrix (28S) for major clades in the Pelagiidae.

Values below the diagonal are minimum pairwise genetic distances computed using Kimura 2-parameter substitution model (Kimura 1980) in MEGA 7.0.14 (Kumar, Stecher & Tamura, 2016). Values in bold represent maximum within clade divergences. Column/row numbers represent major taxa/clades: 1. Chrysaora achlyos, 2. C. africana, 3. C. chesapeakei, 4. Chrysaora c.f. chesapeakei, 5. C. chinensis, 6. C. colorata, 7. C. fulgida, 8. C. fuscescens, 9. C. hysoscella, 10. C. lactea, 11. C. melanaster, 12. C. pacifica, 13. C. plocamia, 14. C. quinquecirrha, 15. Pelagia benovici, 16. P. noctiluca, 17. Sanderia malayensis, 18. Cyanea capillata.

Click here for additional data file.

Supplemental Information 6 Figure S1: Unknown pelagiid jellyfish from Dakar, Senegal.

Photograph of unknown pelagiid jellyfish from Dakar, Senegal ultimately assigned to the species Mawia benovici based on DNA sequence data (28S and COI). The jellyfish bell was approximately 5–6 cm. Photograph courtesy of Lucy Keith-Diagne.

Click here for additional data file.

Supplemental Information 7 Figure S2: Tentacle Nematocyst Sizes.

Average size measurements based on 10 nematocysts per individual (length vs. width) for nematocysts: A) a-isorhizas; B) O-isorhizas; C) heterotrichous microbasic birhopaloids. Error bars represent standard deviation values. Squares represent nematocysts from estuarine Atlantic and Gulf of Mexico medusae (C. chesapeakei), while circles represent nematocysts from coastal Atlantic medusae (C. quinquecirrha). All animals with 16S sequences matching the C. chesapeakei clade appear in red, while those whose sequences matched the C. quinquecirrha clade appear in blue. Triangles represent average values from Papenfuss (1936) for morphs identified as Dactylometra quinquecirrha (gray) or Dactylometra quinquecirrha var. chesapeakei (white). Nematocyst examples are to the right of each graph. All nematocysts are of average size for the nematocyst type and species. Photographs have been resized so that all error bars are the same size on the page to allow size comparisons.

Click here for additional data file.

Supplemental Information 8 Figure S2: Tentacle Nematocyst Diversity.

A) Mosaic plot showing the relative proportions of nematocyst types in distal, medial and proximal tentacle regions. O-isorhiza and birhopaloid nematocysts vary markedly in abundance across regions. Plot drawn using R package vcd (Meyer, Zeileis & Hornik, 2016). Proportions of nematocysts types vary significantly across tentacle regions; shading indicates significant departures from expected values (red = negative residuals, blue = positive residuals). B) Non-metric multidimensional scaling of similarities in overall (proximal, medial and distal regions) proportions of all four nematocyst types. Squares represent nematocysts from estuarine Atlantic and Gulf of Mexico medusae, while circles represent nematocysts from coastal Atlantic medusae. All animals with 16S sequences matching the C. chesapeakei clade appear in red, while those whose sequences matched the C. quinquecirrha clade appear in blue.

Click here for additional data file.

Supplemental Information 9 Raw data from this manuscript included morphological and tentacle nematocyst data.

Click here for additional data file.

We are grateful to John McDonald for his guidance during the developmental phases of the project and his vital manuscript edits. We acknowledge the following for collecting samples or aiding sample collection: Emmanuelle Buecher, Luciano Chiaverano, Mike Davis, Elif Demir, Chris Doller, Tom Doyle, Lisa-Ann Gershwin, Mark Gibbons, Monty Graham, Bill Hall, Shannon Howard, Lucy Keith-Diagne, Monica Martinussen, George Matsumoto, Hermes Mianzan, Wyatt Patry, Jennifer Purcell, Steve Spina, Barbara Sullivan, the crew and personnel of the R/V Cape Henlopen, The Port Royal Marine Laboratory, the Monterey Bay Aquarium, the Aquarium of the Americas and the South Carolina Aquarium. Some molecular and microscopic work was performed using resources of the Laboratory of Analytical Biology at the Smithsonian National Museum of Natural History and some molecular work was performed in the labs of Dr. William Graham (Dauphin Island Sea Lab) and Dr. Michael Dawson (University of California Merced). We acknowledge Scott Whittaker for his micropscopic assistance. We are thankful to Phillipe Bouchet, Dale Calder and Steve Cairns for their critical nomenclatural advice.

Additional Information and Declarations

Competing Interests

Author Contributions

DNA Deposition

The authors declare that they have no competing interests.

Keith M. Bayha conceived and designed the experiments, performed the experiments, analyzed the data, contributed reagents/materials/analysis tools, wrote the paper, prepared figures and/or tables, reviewed drafts of the paper.

Allen G. Collins conceived and designed the experiments, analyzed the data, contributed reagents/materials/analysis tools, wrote the paper, reviewed drafts of the paper.

Patrick M. Gaffney conceived and designed the experiments, analyzed the data, contributed reagents/materials/analysis tools, wrote the paper, reviewed drafts of the paper.

The following information was supplied regarding the deposition of DNA sequences:

MF141552–MF141718; MF167556–MF167568

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
