# Peer review of "Multigene phylogeny of the scyphozoan jellyfish family Pelagiidae reveals that the common U.S. Atlantic sea nettle comprises two distinct species (Chrysaora quinquecirrha and C. chesapeakei)"

_PeerJ, doi:10.7717/peerj.3863_

## Round 0.1 · original submission · Minor Revisions

Both referees agree that the manuscript is interesting, sound and worthy of publication, but each also makes a number of suggestions for minor revisions prior to final acceptance. I do not expect any of these revisions to be difficult to incorporate, and I look forward to seeing your revised manuscript returned.

·

Basic reporting

The text presents an important contribution to the understanding of the diversity of jellyfishes from US east coast, by presenting new and detailed data on a well-known thought to be a single species (until now named Chrysaora quinquecirrha). I am not a native English-speaker, but the manuscript is well written, and the language used is very clear throughout the text. “Introduction” section provides background on the topic and included traditional and updated literature. The structure of the text is in accordance with the standards, but I suggest the authors to move the “Systematics” section before the “Discussion”. Figures and tables are of high quality and essential for helping to clarify the results, unless when noted.

Experimental design

The methods were clearly described and are updated with the available literature.
Conclusions are well supported by the data presented.

Validity of the findings

Although there was an indication in the literature about the occurrence of two varieties of the jellyfish studied, the authors provided robust data to identify and sort them out. The nomenclatural issues were carefully followed and the main result (species name validation) is sustained based on the articles of the Code. Summing up, my evaluation of the manuscript is that it corresponds to an important contribution to Marine Biology in general as it provides advances in the field, especially unravelling the identity of a significant jellyfish species of the Atlantic US coast.

Additional comments

In general the authors were cautious in not proposing new names for the different groups found among the genus Chrysaora. I think that additional data must be gathered to perform such big changes in the classification of the family. There are some minor corrections and suggestions performed directly on the PDF file, summarized as follows in order of importance: 1) holotype specimen must be deposited in a well-known collection, as well as other comparison individuals; 2) provide further data on the Senegalese specimens (images as supplementary material would be welcome); 3) correct some references regarding format and some data.

·

Basic reporting

No Comment

Experimental design

No comment

Validity of the findings

No Comment

Additional comments

PeerJ
Multigene phylogeny of the scyphozoan jellyfish
family Pelagiidae reveals that the common U.S.
Atlantic sea nettle comprises two distinct species
(Chrysaora quinquecirrha and C. chesapeakei)
(#17919)

Brief summary

This study seeks to reconstruct the phylogenetic relationships among 4 pelagiid genera and 11 of 13 species of Chrysaora. In addition collections were made along the US Atlantic and Gulf coast of C. quinquicirrha, from which molecular, gross anatomical and cnidomic data were collected. The stated goals of the authors was to clarify taxonomic status among genera and species of this group of nuisance species. The molecular approach used nuclear 28S and mtDNA COI and 16S. Morphological analyses focused on tentacle and lappet number, oral arm length and nematocyst dimensions. Results suggest that C. quinquicirrha is polyphyletic, and comprises two distinct lineages one from the Gulf and the other from the Atlantic. This conclusion is supported by molecular as well as morphological evidence.


This paper is certainly interesting scientifically as well as provides an important contribution to applied resource management. The phylogenetic analysis presented is technically sound, and the resulting systematic claims are convincing. Figures are of sufficient quality, including maps, photos and trees.

The systematics sections is about 11 pages, raising the editorial question of whether this section is appropriate for the journal PeerJ, or might be included as an appendix, otherwise this could be more appropriate for a taxonomic journal.

I have made a number of minor grammatical suggestions and minor corrections of typos.



Specific Comments
L 122
Please italicize “Sanderia”

L153
Use of the small unprimed italicized letter implies that the heme prosthetic group is in a hemochrome linkage, and a lower-case italicized letter, e.g., c', should be used.
L184
Italicize “c”

L201 and L203
Please write out the NIH DNA database as “GenBank”

L219
“jModelTest”

L328
I’d suggest rewording “two distinct clades that were highly diverged” “two distinct, highly divergent clades”

L332 & 335 and throughout
Please be consistent: “Figs” vs. “Figures”

L337
Replace “highly-supported” with “well-supported”

L344
Replace “observed” with more appropriate term, such as “evaluated” or “examined”

L345-6
“were observed for 20 quantitative and qualitative macromorphological characters either taken from Gershwin and Collins (2002) or new to this study (maximum oral arm length)”

This sentence mentions 20 traits, and lists a single trait that was unique to this study. It remains unclear to the reader why a single character is listed, this could lead to confusion as to whether this single trait listed is the only trait used that was not one from the Gershwin and Collins 2002 paper, whereas if those traits that fall into both categories are indicated in a table, perhaps simply referring to the table would be more clear at this point.

L361
In: “<~1.5%” inclusion of the symbol “~” seems awkward, does removing this symbol drastically alter the meaning? If not I’d suggest deleting it.

L374
“all those sequenced from” this is admittedly a bit of a pet peeve of mine, but to state that “individuals” are sequenced is inaccurate, strikes me a slang, since DNA, or gene fragments are sequenced rather than jellyfish, or individuals being sequenced. I’d suggest rewording to something like “jellyfish included in the molecular analysis” or something along these lines that more precisely reflects the author’s intended meaning.

L423
Replace “a paraphyletic Chrysaora.” with “paraphyly of Chrysaora.”
L434
Should a question mark be added following “limited”

L475
“therefore, conclusions based on these sequences should be made with care.” Why? Perhaps consider explaining, for instance “because the geographic provenance of these individuals is unclear”, or what ever the reasoning.

L484
The following sentence: “C. quinquecirrha occurred in two well-differentiated” opens with an abbreviation of the genus, whereas starting a sentence with an abbreviation should be avoided. This is an editorial, and grammatical decision.

L596
“and” not italicized

L604 and 608
Include “(MBA)” after first mention of “Monterey Bay Aquarium”.

L634
“(Figure 4 and 6)” please follow journal guidelines and be consistent in the manner in which two figures are referred to, there are instances where the following is used “Figures 4, 6”

---

## Round 0.2 · accepted · Accept

Having read through the revisions and your responses to the referees, I am happy to move your article forward into press.